# Inhibition of neuronal FLT3 receptor tyrosine kinase alleviates peripheral neuropathic pain in mice

Cyril Rivat[1,2], Chamroeun Sar[1,2], Ilana Mechaly[1,2], Jean-Philippe Leyris[1,3], Lucie Diouloufet[1], Corinne Sonrier[1,3], Yann Philipson[4], Olivier Lucas[1], Sylvie Mallié[1,2], Antoine Jouvenel[1,2], Adrien Tassou[1,2], Henri Haton[1,2], Stéphanie Venteo[1], Jean-Philippe Pin [5], Eric Trinquet[6], Fabienne Charrier-Savournin[6], Alexandre Mezghrani[1], Willy Joly[1], Julie Mion[1], Martine Schmitt[4], Alexandre Pattyn[1], Frédéric Marmigère[1], Pierre Sokoloff [3], Patrick Carroll [1], Didier Rognan[4] & Jean Valmier[1,2]

Peripheral neuropathic pain (PNP) is a debilitating and intractable chronic disease, for which sensitization of somatosensory neurons present in dorsal root ganglia that project to the dorsal spinal cord is a key physiopathological process. Here, we show that hematopoietic cells present at the nerve injury site express the cytokine FL, the ligand of fms-like tyrosine kinase 3 receptor (FLT3). FLT3 activation by intra-sciatic nerve injection of FL is sufficient to produce pain hypersensitivity, activate PNP-associated gene expression and generate short-term and long-term sensitization of sensory neurons. Nerve injury-induced PNP symptoms and associated-molecular changes were strongly altered in *Flt3*-deficient mice or reversed after neuronal FLT3 downregulation in wild-type mice. A first-in-class FLT3 negative allosteric modulator, discovered by structure-based in silico screening, strongly reduced nerve injury-induced sensory hypersensitivity, but had no effect on nociception in non-injured animals. Collectively, our data suggest a new and specific therapeutic approach for PNP.

[1] Institute for Neurosciences of Montpellier, INSERM, Institut National de la Santé et de la Recherche Médicale, UMR1051, Hôpital Saint-Eloi, Montpellier 34000, France. [2] Université de Montpellier, Montpellier 34000, France. [3] Biodol Therapeutics, Cap Alpha, Clapiers 34830, France. [4] Laboratoire d'Innovation Thérapeutique, UMR7200, CNRS-Université de Strasbourg, Illkirch 67400, France. [5] Institut de Génomique Fonctionnelle, CNRS, INSERM, Univ. Montpellier, 34094 Montpellier, France. [6] Cisbio Bioassays, Parc Marcel Boiteux, BP8417530200 Codolet, France. These authors contributed equally: Cyril Rivat, Chamroeun Sar, Ilana Mechaly, Jean-Philippe Leyris. Correspondence and requests for materials should be addressed to D.R. (email: rognan@unistra.fr) or to J.V. (email: jean.valmier@umontpellier.fr)

Peripheral neuropathic pain (PNP), for which specific and effective therapies are lacking, is a broad public health problem particularly due to its high prevalence (estimated to be 6.9–10% in the general population[1]), debilitating effects, and its high social cost[2,3]. Currently, PNP treatments consist essentially in repurposed anti-depressant drugs (e.g., tricyclic anti-depressants and serotonin-noradrenaline uptake inhibitors) and anti-epileptic drugs of the class of $\alpha 2\delta - 1$ voltage-gated calcium channels blockers, both having poor efficacy and producing several side effects[4]. There is a crucial need for specific PNP medications, which requires the identification of new specific targets implicated in the initiation and maintenance of PNP.

PNP arises from aberrant functioning of somatosensory neurons present in dorsal root ganglia (DRG) that project to the dorsal spinal cord (DSC)[5,6]. Environmental stimuli are converted into voltage changes in somatosensory neurons by ionic transducer channels that respond to specific thermal, mechanical, and chemical stimuli and activate sodium channels that generate and propagate action potentials to the DSC. Nerve injury rapidly induces peripheral sensitization due to reduced thresholds of both transducers and voltage-activated channels, increasing responsiveness to stimuli, and axonal hyperexcitability[7,8]. Within hours, multiple adaptive modifications occur in the DRG, including gene expression changes, post-translational protein alterations, and modifications of protein trafficking. For example, after nerve damage, TRPV1, a member of the TRP transducer family[9], with a well-established role in inflammatory pain, is upregulated in different models of PNP both at DRG peripheral and central synapses. This upregulation is correlated with the development and the maintenance of thermal hypersensitivity[10]. In addition, decreasing TRPV1 levels or inhibiting its activity reduces part of

the neuropathic hypersensitivity[11,12]. Peripheral sensitization leads to central sensitization in the DSC that is the cornerstone of PNP chronification. Currently, how peripheral sensitization develops and persists is incompletely understood.

Neuro-immune interactions are key regulators of local peripheral sensitization[13,14]. They are mediated by immune cells, which invade the lesion site after nerve blood barrier permeabilization, secrete sensitizers (cytokines, chemokines, growth factors) that contribute to the development of the nerve injury-induced hypersensitivity and the maintenance of PNP[13,14]. Cytokines and their receptors have been identified as important actors in these interactions[15]. Among them, with the notable exception of the cytokine FL[16,17] and its cognate receptor FLT3[18], all the members of the class III receptor tyrosine kinase (RTK) family, which comprises stem cell factor (SCF) receptor (c-Kit), colony-stimulating factor type-I (CSF1) receptor (CSF1R), and platelet-derived growth factor (PDGF) receptors (PDGFR) have been shown to be involved in normal nociception and/or pain[19–22]. For example, nerve injury induces de novo neuronal expression of CSF1, which is transported to the DSC where it targets CSF1R expressed by microglia cells[19]. Microglia activation is considered as a major factor in central PNP chronification[13,14]. However, the mechanism leading to CSF1 induction is unidentified.

FLT3 is expressed in most hematopoietic organs, such as spleen, thymus, peripheral blood and bone marrow, and its gain-of-function mutation promotes hematopoietic cells proliferation that is targeted for therapy of hematologic cancers[23]. However, $Flt3^{KO}$ mice have a normal number of mature hematopoietic cells in all hematopoietic organs, even though some discrete populations of cell progenitors, but not more mature cells, are reduced[24].

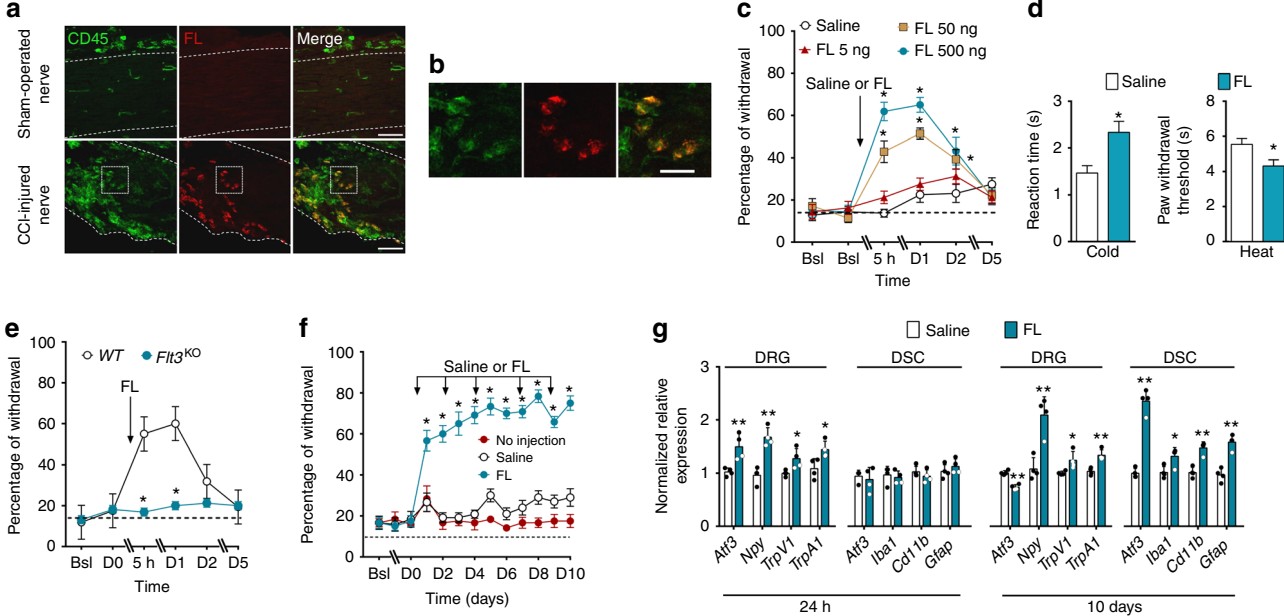

**Fig. 1** Infiltration of CCI-injured nerve by CD45/FL-positive cells and effects of intrathecal FL injections on pain sensitivity and PNP-related biomarkers. **a** CD45- and FL-positive cells at the site of nerve injury in the sciatic nerve, delineated by dotted lines, 10 days post-CCI. Scale bars = 50 μm. **b** Enlargement of the dotted square areas in **a**. Bars = 20 μm. **c** Mechanical pain sensitivity, measured as the percentage of withdrawal to application of von Frey filament, 5 h (5 h) to 5 days (D5) after a single intrathecal saline or different doses of FL injection (concentrations in ng/10 μl). Bsl basal score before injection. **d** Cold and heat thermal sensitivity 24 h after intrathecal saline or FL injection (500 ng). **e** Mechanical sensitivity after a single intrathecal injection of FL (500 ng/10 μl) in wild-type (WT) or Flt3-deficient ($Flt3^{KO}$) mice. **f** Mechanical pain sensitivity after repeated intrathecal injections of saline or FL at the dose of (500 ng/10 μl), measured 1–10 days after FL injections performed on days 1, 2, 4, 7, and 9. Control animals received no injections. **g** Normalized gene expressions of PNP biomarkers in the DRG and DSC measured by q-PCR 24 h after a single saline or FL injection (500 ng/10 μl) and 10 days after repeated injections as in **f**. Results are means ± s.e.m. of data from 8 animals **c–f** or 4 animals (**g**). Two-way ANOVA and Dunnett's test (**c, e, f**). Unpaired Student's t-test (**d, g**).*P < 0.05; **P < 0.01 vs. saline or WT or no injection

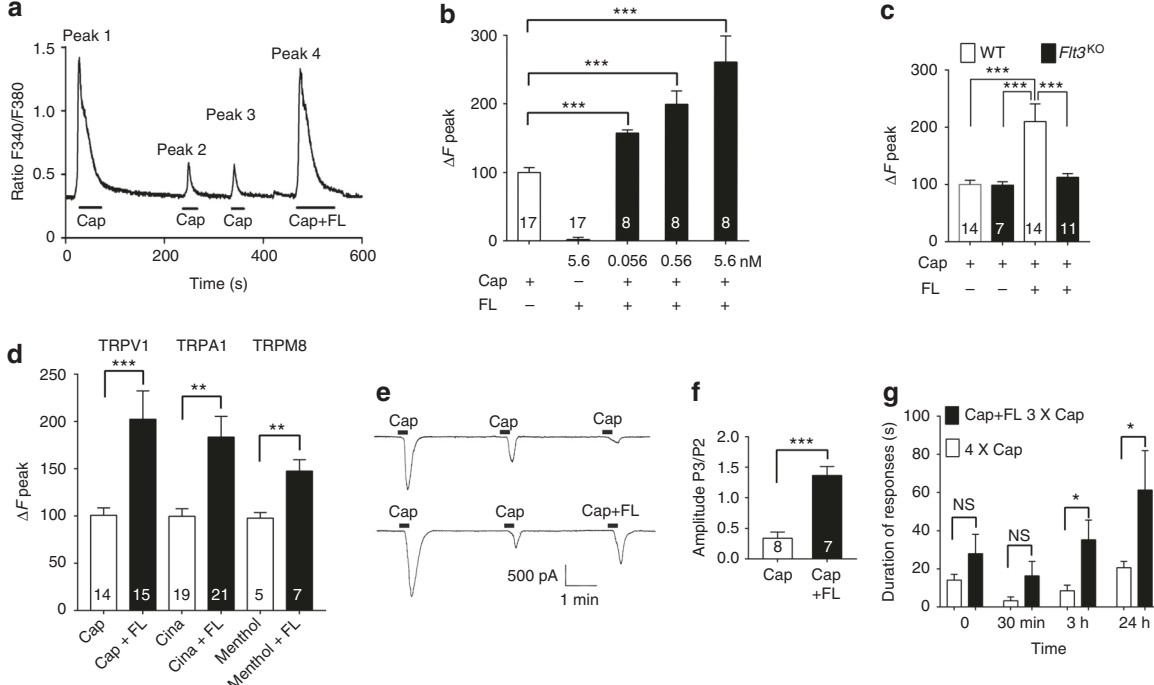

**Fig. 2** FL potentiates TRP function in vitro and in vivo. **a** Traces of $[Ca^{2+}]_i$ responses to repeated bath applications of capsaicin (Cap, 2.5 μM), a selective TRPV1 activator, or capsaicin combined with FL (0.56 nM) on $[Ca^{2+}]_i$ levels measured by real-time $Ca^{2+}$ cell imaging in cultured DRG neurons. Ratio F340/F380 is the ratio of fluorescence signals measured at 340 and 380 nm excitation wavelengths. **b** Effects of capsaicin (Cap, 2.5 μM) or FL at the indicated concentrations, either alone or in combinations on $[Ca^{2+}]_i$ levels, measured at the fourth application of capsaicin as in **a**. Results are expressed as response amplitudes normalized to capsaicin alone ($\Delta F$ peak). **c** Effects of FL on capsaicin-induced increased $[Ca^{2+}]_i$ levels, measured as in **a**, in DRG neurons from wild-type (WT) or $Flt3^{KO}$ mice. **d** Potentiation by FL (0.56 nM) of increase in $[Ca^{2+}]_i$ levels induced by capsaicin (Cap, 2.5 μM), cinnamaldehyde, Cina (50 μM), and menthol (1 μM). **e** Traces of voltage-clamp whole-cell recording of capsaicin (Cap, 1 μM)-induced TRPV1 currents and their potentiation by FL (10 nM). **f** Quantification of FL effects measured as in **e**. **g** In vivo potentiation by FL of capsaicin-induced pain-related behaviors. Animals received either 4 repeated identical injections of capsaicin (15 ng) or a combination of Cap and FL (2 ng) followed by 3 repeated capsaicin injections in the paw, at the indicated times. Results are means ± s.e.m. of data from the number of cells indicated in columns (**b**–**d**, **f**) or data from 8 animals (**g**). Unpaired Student t-test. NS non-significant; *$P < 0.05$; **$P < 0.01$; ***$P < 0.001$

$Flt3$ transcripts are expressed in various regions of the nervous system, in particular in human and mice DRGs[25–28]. FLT3 modulates the in vitro survival of embryonic mice DRG neurons[28] but its role in the adult nervous system is not known. Here we demonstrate that neuronal FLT3 in DRG is a critical actor for PNP initiation and maintenance in different mouse models. In addition, we show that PNP can be alleviated by specific FLT3 inhibition using a new chemical entity, thus identifying a novel therapeutic strategy for PNP treatment.

## Results

**FL induces PNP-related behavioral and molecular changes**. We used the chronic constriction injury (CCI) model in mice as a model of persistent PNP, consisting in three chronic ligatures tied loosely around the sciatic nerve[29]. On longitudinal sections of sciatic nerve 10 days after nerve injury, intense FL immunoreactivity was present in the nerve at the site of injury in CCI mice, whereas it was absent in normal, sham-operated nerve. All of the FL-positive cells also expressed CD45, a marker of the immune hematopoietic lineage[30] (Fig. 1a, b; Supplementary Fig. 1a). Further analysis showed that 60% of the FL-positive cell population express CD11b, a marker of myeloid cells, but not CD68, a marker of macrophages (Supplementary Fig. 1b, c). No co-localization was found with CD3, a marker of T lymphocytes (Supplementary Fig. 1d). Therefore, FL-expressing cells that penetrate the nerve at the lesion site can be identified as monocytes, neutrophils, and/or natural killer cells but not macrophages or T lymphocytes.

To assess whether FL could participate in the generation of PNP symptoms, uninjured mice were injected intrathecally with recombinant FL and their sensitivity to mechanical stimulation of the hind-paw was measured. A single FL injection induced a dose-dependent increase in the percentage of paw withdrawal to a calibrated von Frey filament (Fig. 1c), i.e., mechanical hypersensitivity, a hallmark of PNP[31], which was present 5 h post-injection and persisted for at least 2 days. In a separate experiment, the onset of FL-induced mechanical pain hypersensitivity was determined to be 120 min: the percentage of paw withdrawal after an intrathecal injection of 500 ng of FL were (means ± s.e.m., $n = 8$) 31.3 ± 4.5, 18.8 ± 2.6 and 51.3 ± 3.2% at 30, 90 and 120 min post-FL injection, respectively, whereas those after saline injection were 30.0 ± 3.7, 21.3 ± 3.4, and 21.3 ± 3.9% ($P < 0.0001$ for comparison of FL vs. saline at 120 min by two-way ANOVA, not significant at other time points).

FL-injected mice also displayed thermal hypersensitivity in response to cold and hot stimuli (Fig. 1d). Mice with a homozygous deletion of $Flt3$ ($Flt3^{KO}$ mice) failed to develop mechanical hypersensitivity following FL injection (Fig. 1e), whereas they displayed normal proportions of sensory neurons expressing typical molecular markers, motor behaviors, and pain responses to nociceptive stimuli (Supplementary Fig. 2a, b). This indicates that FL-induced hypersensitivity resulted from activation of its cognate FLT3 receptor. Repeated injections of FL every 2 days during 10 days maintained mechanical hypersensitivity that persisted as long as the treatment continued (Fig. 1f).

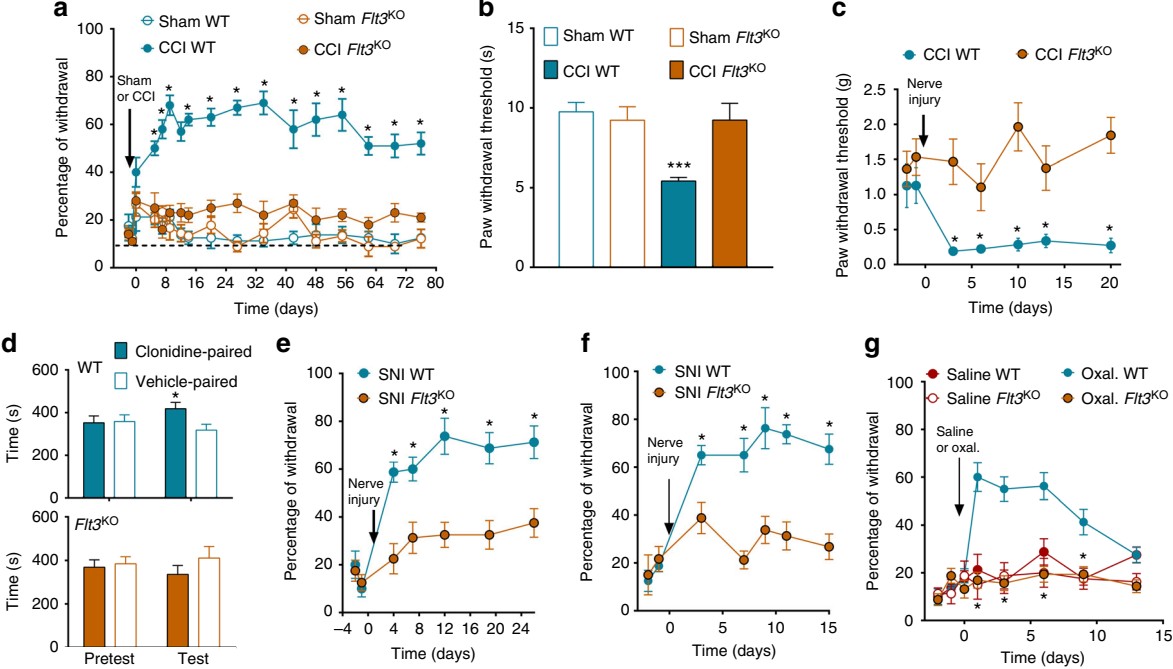

**Fig. 3** FLT3 is critical for the development of pain hypersensitivity. **a–g** Flt3-knockout (Flt3[KO]), compared to wild-type (WT) animals, showed marked reductions of hypersensitivity to mechanical nociceptive stimuli as measured by the von Frey test (**a**), thermal stimuli as measured via the Hargreaves test 14 days post-injury (**b**) and punctate tactile stimuli (**c**), and of conditioned place preference induced by clonidine-evoked analgesia 14 days post-injury (**d**). Flt3[KO] mice also showed decrease of mechanical pain hypersensitivity in the spared nerve injury (SNI) model (**e**), in the sciatic nerve ligature (SNL) model (**f**) and in the oxaliplatin model of generalized peripheral neuropathic pain (**g**). All the values are means ± s.e.m. ($n = 8$ except in **d**, $n = 11$). Two-way ANOVA and Dunnett's test (**a–c**, **f**); Student's $t$-test (**d**). Two-way ANOVA and Bonferroni's test (**e**, **g**), *$P < 0.05$; ***$P < 0.001$ vs. Sham WT, Sham CCI, CCI Flt3[KO], or vehicle paired

We next examined changes in nerve injury-associated genes in DRG and DSC. Twenty-four hours after a single FL injection, expression levels in DRG of the stress-induced gene transcript Atf3[32] and several important neuronal pain-related gene transcripts, e.g., neuropeptide Y (NpY)[33] and transient receptor potentials TrpV1[34] and TrpA1[35] were increased (Fig. 1g). Expression levels of PNP-associated gene transcripts in the DSC at this time point (24 h) showed no change compared to saline-injected animals (Fig. 1g). In contrast, repeated injections of FL over 10 days caused striking changes in the DSC expression of genes associated with the process of central pain sensitization[36], notably the activated microglia markers Iba1 and Cd11b[37] and the activated astrocyte marker Gfap[38] (Fig. 1g). Thus, FLT3 activation causes, in the short-term, upregulation of PNP-related genes in the DRG, and in the long-term, molecular changes in the DSC typical of those occurring during chronification of PNP.

In cultured DRG neurons, application of capsaicin, a specific TRPV1 agonist, increased intracellular $Ca^{2+}$ levels ($[Ca^{2+}]_i$) and repeated capsaicin applications attenuated TRPV1 responses, which reflects receptor desensitization (Fig. 2a)[39]. FL alone had no effect on basal $[Ca^{2+}]_i$ (Fig. 2b), but markedly potentiated, in a concentration-dependent manner, the TRPV1 response to repeated capsaicin applications (Fig. 2b). Although the capsaicin response was normal in cultures established from Flt3[KO] mice, the potentiating effect of FL was completely abolished (Fig. 2c). Similarly to its effect on TRPV1 function, FL potentiated both the $[Ca^{2+}]_i$ response to cinnamaldehyde and menthol, specific activators of TRPA1 and of TRPM8 channels, respectively (Fig. 2d). Voltage-clamp whole-cell recording confirmed that capsaicin-induced TRPV1 currents were effectively potentiated by FLT3 activation (Fig. 2e, f). Furthermore, hind-paw FL injection also potentiated capsaicin-induced spontaneous pain-related behaviors (Fig. 2g).

Thus, acute and chronic in vivo FLT3 activation by FL recapitulates in mice some of the molecular and functional changes in sensory neurons and behavioral alterations that are normally induced by peripheral nerve injury. Altogether, these data strongly suggest that FL directly acts on primary sensory neurons. Indeed, injection of FL directly into the sciatic nerve, but not systemic FL augmentation by intravenous injection (Supplementary Fig. 3a, b), caused mechanical hypersensitivity, showing that systemic peripheral activation of FLT3 is not involved in pain behavior. This indicates that FLT3 triggering PNP-like symptoms is present in the nerve and/or DRG. In agreement with this observation, Western blot detected FLT3 in DRG tissue from WT, but not Flt3[KO] mice, (Supplementary Fig. 3c) and Flt3 mRNA was visualized in DRG neurons by in situ hybridization (Supplementary Fig. 3d).

**Neuronal FLT3 controls PNP development and maintenance.** Considering the similarities in the effects of FLT3 activation and those induced by peripheral nerve injury, we then asked whether downregulation of FLT3 functioning could influence the molecular, cellular, and behavioral responses to nerve injury. In the CCI model, injury of the sciatic nerve produced mechanical hypersensitivity that lasted for more than 2 months in WT mice, whereas Flt3[KO] mice failed to develop pain-related behavior over the same period (Fig. 3a). Note that the repetition of painful mechanical stimulus applications is unlikely to produce sensitization or tolerance since the maximal change in paw withdrawal threshold was achieved after 2 weeks and was maintained over time until the end of the experiment. The heat hypersensitivity and mechanical allodynia observed in WT mice after CCI were also absent in Flt3[KO] mice (Fig. 3b, c). The conditioned place preference paradigm has been used to reveal the presence of non-

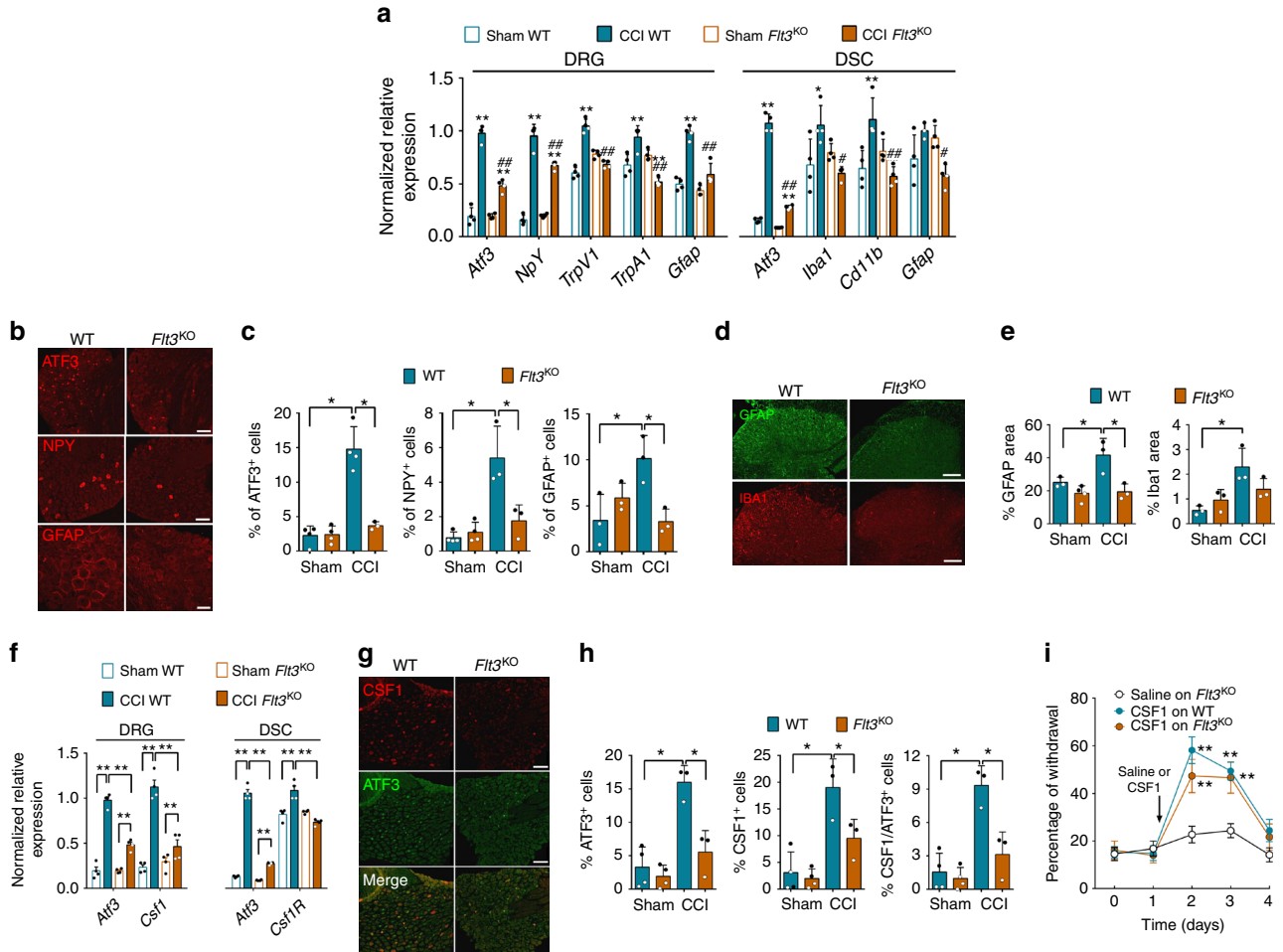

**Fig. 4** FLT3 is critical for upregulation of markers associated with peripheral nerve injury. **a** Changes in PNP-related biomarkers mRNA expression, measured by q-PCR 3 days after sham or CCI surgery in wild-type (WT) or *Flt3*-knockout (*Flt3*KO) mice. Values are means ± s.e.m of data from 4 animals. One-way ANOVA and Bonferroni's test, *$P < 0.05$; **$P < 0.01$ vs. respective Sham, #$P < 0.05$; ##$P < 0.01$ vs. CCI WT. **b** ATF3, NPY, and GFAP immunoreactivity in DRG from WT and *Flt3*KO CCI mice, 14 days post-CCI. Scale bars = 100 μm except for GFAP: 50 μm. **c** Quantification of experiments as in **b**. **d** GFAP and IBA1 immunoreactivity in DSC from WT and *Flt3*KO CCI mice. Scale bars = 100 μm. **e** Quantification of experiments as in **d**. **f** Normalized gene expressions of *Csf1* and *Csf1R* measured by q-PCR in DRG and DSC from WT and *Flt3*KO sham or 3 days post-CCI mice. Means ± s.e.m. of data from 4 animals. One-way ANOVA and Bonferroni's test, *$P < 0.05$; **$P < 0.01$ vs. respective Sham, #$P < 0.05$; ##$P < 0.01$ vs. CCI WT. **g** CSF1 and ATF3 immunoreactivity in DRG from WT and *Flt3*KO CCI mice at 3 days post-injury. Bars = 100 μm. **h** Quantification of experiments as in **g**. **i** CSF1-induced mechanical pain hypersensitivity in WT and *Flt3*KO mice. Means ± s.e.m. of data from 8 animals. **c**, **e**, **f**, **h**, **i** NS non-significant; unpaired Mann–Whitney *t*-test (**c**, **d**, **f**, **h**); two-way ANOVA and Bonferroni's test (**i**), *$P < 0.05$; **$P < 0.01$ vs. saline

evoked ongoing pain in nonverbal animals[40]. Administration of the non-rewarding and rapidly-acting analgesic drug clonidine, referred to as clonidine-induced analgesia, produced a place preference in wild-type CCI animals, which was totally abolished in *Flt3*KO CCI mice (Fig. 3d), suggesting a loss of non-evoked ongoing pain in *Flt3*KO mice. Furthermore, the results on nerve injury-induced mechanical hypersensitivity could be extended to different PNP models, the spared nerve injury (SNI)[41] and spinal nerve ligation (SNL)[42] models, and also to the oxaliplatin model of chemotherapy-induced generalized PNP (Fig. 3e–g). Nevertheless, *Flt3*KO mice have normal responses to chemically-induced nociception in the formalin test and in the complete Freund's adjuvant (CFA) chronic inflammatory pain model (Supplementary Fig. 4a, b).

We next assessed whether the deletion of *Flt3* could modulate the expression of pain-related factors induced by nerve injury involved in peripheral and central sensitization responsible for PNP chronification[14]. The injury-induced increases in transcripts of *Atf3*, *NpY*, *TrpV1*, and *Gfap* in DRG and *Iba1*, *Cd11b* in DSC

were abrogated or diminished after *Flt3* deletion, at 3 days post-CCI (Fig. 4a). These variations were confirmed by immunochemistry analysis, showing that, in DRG, *Flt3* deletion inhibited the injury-induced increases in ATF3, NPY, and GFAP, seen in WT mice at 14 days post-CCI (Fig. 4b, c). In DSC, *Flt3* deletion also diminished nerve injury-induced changes of astrocyte (GFAP) and a similar trend was seen for microglial (IBA1) activation markers (Fig. 4d, e). *Flt3* deletion in CCI mice attenuated the upregulation of *Csf1* ligand transcript in DRG and *Csf1R* in DSC (Fig. 4f). These results were confirmed by the quantification of reduced numbers of DRG CSF1-positive cells using immunohistochemistry (Fig. 4g, h). Finally, intrathecal CSF1 injection induced mechanical hypersensitivity in both WT and *Flt3*KO mice (Fig. 4i), suggesting that FLT3 acts upstream of CSF1/CSF1R, a major signaling pathway in neuropathic pain chronification[19].

**Downregulation of *Flt3* expression reverses established PNP.** To determine whether PNP-like symptoms, once established, could be reversed by *Flt3* downregulation, we injected

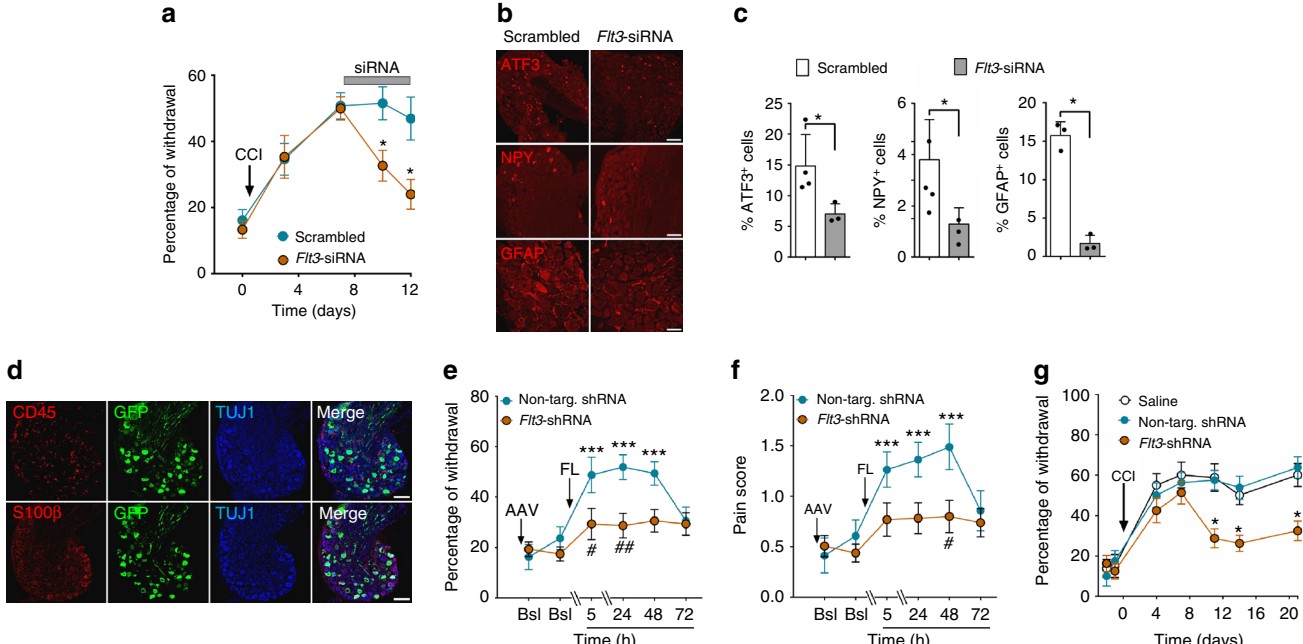

**Fig. 5** Curative effects of *Flt3* downregulation on PNP-related genes, and nerve injury-induced and FL-induced pain hypersensitivity. **a** Mechanical pain hypersensitivity after intrathecal infusion for 4 days of scrambled- or *Flt3*-siRNA in CCI mice. Means ± s.e.m. of data from 8 animals. Two-way ANOVA and Dunnett's test, *$P < 0.05$ vs. Scrambled siRNA animals. **b** ATF3, NPY, and GFAP immunoreactivity in lumbar DRG after intrathecal infusion for 4 days of scrambled- or *Flt3*-siRNA in CCI mice. Bars = 100 μm except for GFAP: 50 μm. **c** Quantification of experiments as in **b**. Means ± s.e.m. of data from 4 animals. Mann–Whitney *t*-test, *$P < 0.05$. **d** Localization of the AAV9 virus expressing anti-*Flt3* shRNA and GFP in neurons (TUJ1-positive cells), but not in CD45- or S100β-positive cells in DRG of WT mice. Scale bars = 100 μm. **e**, **f** Effects of *Flt3*-shRNA on FL-induced mechanical pain hypersensitivity (**e**), and on pain-related behaviors (**f**). FL (500 ng/10 μl) was injected intrathecally 17 days after the virus. Means ± s.e.m. of data from 8 animals. One-way ANOVA and Dunnett's test, *$P < 0.05$; **$P < 0.01$; ***$P < 0.001$ vs. Bsl; two-way ANOVA with repeated measures and Dunnett's test, #$P < 0.05$ vs. non-targeting shRNA. **g** Effects of *Flt3*-shRNA on mechanical pain hypersensitivity in CCI mice. The virus was injected intrathecally 48 h before CCI. Means ± s.e.m. of data from 8 animals. Two-way ANOVA with repeated measures and Dunnett's test, *$P < 0.05$ vs. non-targeting shRNA

intrathecally *Flt3*-directed siRNA in CCI mice via mini-pumps during a 6-day period starting 8 days after the nerve injury. *Flt3*-siRNA (but not scrambled siRNA) effectively down-regulated *Flt3* expression in vitro and FLT3 function ex vivo (Supplementary Fig. 5a–c), as well as CCI-induced mechanical hypersensitivity (Fig. 5a), without affecting normal mechanical nociception (Supplementary Fig. 5d). These changes were accompanied by reductions of PNP-related mRNAs (Supplementary Fig. 5e) and protein levels, as shown by reductions in numbers of cells expressing ATF3, NPY, and GFAP in the DRG (Fig. 5b, c). To determine whether neuronal FLT3 in DRG is necessary and sufficient to regulate PNP symptoms, we constructed an AAV9 virus vector co-expressing an *Flt3* shRNA (*Flt3*-sh) and the green fluorescence protein (GFP). When injected intrathecally, virus-derived GFP expression in DRG was restricted to neurons (Fig. 5d) and in DSC appeared only in fiber-like processes, most likely originating from sensory neuron projections. GFP expression was not present in DSC cell bodies (Supplementary 6a, b). Following reduction of FLT3 protein levels in the DRG after treatment with the *Flt3*-directed shRNA-expressing AAV9 virus (Supplementary Fig. 6c, d), intrathecal FL injections failed to produce evoked mechanical pain hypersensitivity (Fig. 5e), as well as increase pain score, which takes into account pain-related behaviors (Fig. 5f), thus showing that the effect of FL is exerted directly via neuronal FLT3. Furthermore, *Flt3*-directed shRNA largely reduced percentage of withdrawal in CCI mice, whereas an AAV9 virus expressing a non-targeting shRNA had no effect (Fig. 5g). Thus, inhibition of *Flt3* expression reduced mechanical pain hypersensitivity produced by either FL injection or nerve injury. Altogether, these results show that PNP symptoms

induced by peripheral nerve injury are mediated by FLT3 expressed in DRG neurons and that PNP, once established, can be reversed by acute downregulation of *Flt3* gene expression.

**BDT001, an FLT3 inhibitor, reverses PNP symptoms**. The X-ray structure of human FL-FLT3 complex[23,43] (Fig. 6a) was used to design a pharmacophore recapitulating all molecular interactions between the FLT3 D3 domain and the N-terminal part (H8-S13) of the FL ligand. Owing to the surprisingly compact FL-binding epitope (FL-FLT3 interface area of 749 Å²), the FL–FLT3 interaction pharmacophore (Fig. 6b) is simple enough to be fitted by small molecular weight compounds. In silico screening of ca. 3 million commercially available compounds afforded 221 unique compounds fulfilling the pharmacophore, out of which 28 chemically representative hits (Fig. 6c; Supplementary Table 1) were selected for purchase.

One compound (compound Hit #**3**, Fig. 6d, e) effectively prevented extracellular FL binding to FLT3, as measured by time-resolved fluorescence resonance energy transfer (trFRET, see Supplementary Fig. 7a, b for binding assay validation). Among the 21 commercial and close structural analogs of compound **3** (Supplementary Table 2), seven compounds were more potent than parent compound **3** in the competition binding assay (Supplementary Fig. 8a). Notably compounds **66** and **75** inhibited FL binding to FLT3 with IC$_{50}$ values of 11 and 17 μM, respectively (Supplementary Fig. 8a). Binding affinities of the 21 analogs permitted to establish preliminary structure–activity relationships (SAR) on the chemical series (Supplementary Fig. 8b). After in-house synthesis (Supplementary Fig. 9a),

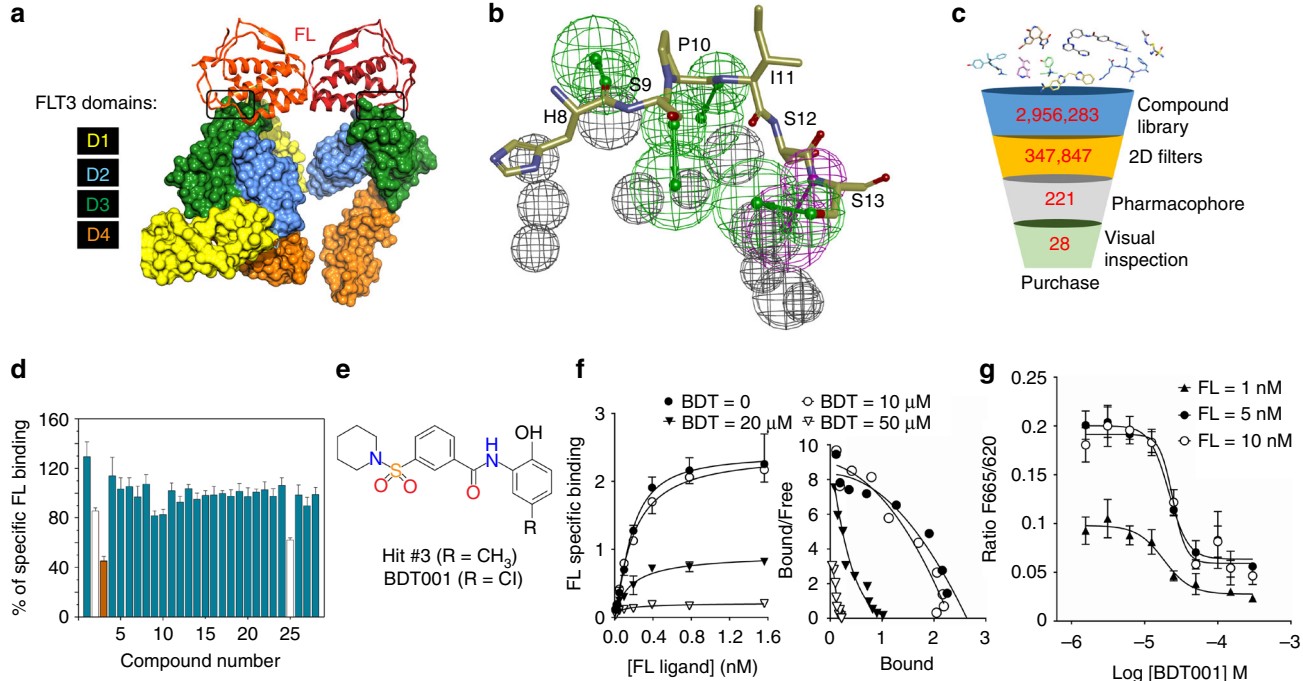

**Fig. 6** Identification of BDT001 as an extracellular FLT3 inhibitor. **a** X-ray structure of human FL–FLT3 complex. The FL dimer is represented by ribbons (monomer 1, firebrick; monomer 2, redorange) and the FLT3 extracellular D1-D4 domains by solid surfaces (D1, yellow; D2, blue; D3, green; D4, orange). The region delineated in black represents the targeted FL/FLT3 interface. **b** FL–FLT3 interaction pharmacophore, comprising four hydrogen-bond acceptors (green), one hydrogen-bond donor (magenta) and 12 exclusion spheres (gray) placed at FLT3 key side chain atoms. **c** Virtual screening flow-chart. Three million commercially available compounds were first filtered according to simple molecular counts and 2D properties and converted into 3D structures (Supplementary Methods). Remaining 347,847 compounds were fitted to the pharmacophore and the best 221 hits selected. Overall, 28 representative chemically diverse hits were finally selected after clustering and visual inspection. **d** Hit experimental validation by inhibition of Red-FL binding to Lumi4-Tb-SNAP-FLT3 in HEK293 cells. Compounds were tested at 100 μM. Compounds #**2** and #**25** altered the fluorescence emission of the donor at 620 nm. Results are means ± s.e.m. of quadruplicate determinations in a single experiment. **e** Chemical structure of hit #**3** and BDT001. **f** Saturation curves of FL binding in the presence of BDT001 (BDT) at the indicated concentrations (left) and Scatchard plots showing non-competitive inhibition and positive cooperativity of FL binding, the latter was lost in the presence of BDT001 at 20 and 50 μM (right). **g** BDT001 inhibited FL-induced FLT3 auto-phosphorylation in RS4–11 cells with $IC_{50}$ values of 18, 24, and 22 μM at FL concentrations of 1, 5, and 10 nM, respectively. In **f**, **g** results are means ± s.e.m. of quadruplicate determinations in a single experiment, which was repeated once with similar results

compound **66** (hereafter referred to as BDT001, Fig. 6e) was confirmed as a true FLT3 inhibitor, which inhibited FL binding in a non-competitive manner and disrupted positive cooperativity of FL binding (Fig. 6f). BDT001 inhibited FL-induced FLT3 phosphorylation in leukemia-derived RS4–11 cells, also measured by trFRET (see Supplementary Fig. 7a, b for phosphorylation assay validation) with an $IC_{50}$ of 18–24 μM that was almost unchanged with increasing FL concentrations (Fig. 6g). These results suggest that BDT001 is an FLT3 negative allosteric modulator.

In primary cultures of adult DRG neurons, BDT001 affected neither capsaicin-induced TRPV1 activation (Fig. 7a) nor basal $[Ca^{2+}]_i$ (Fig. 7b), but reversed in a dose-dependent manner the potentiation by FL, with a maximal effect at 1 μM, and a partial effect at 0.1 μM (Fig. 7b). Thus, the effective BDT001 concentration required for functional inhibition of FLT3 in neurons was significantly lower than that required for inhibiting FL binding and FL-induced FLT3 auto-phosphorylation in RS4–11 cells. Among the rare examples of extracellular RTK inhibitors, two similar situations have already been encountered (Supplementary Note 1). The effect of BDT001 was FLT3-dependent, as the compound at 10 μM had no effect when TRPV1 potentiation was induced by NGF (Fig. 7c).

After systemic administration of BDT001 (5 mg/kg i.p.) in mice, FL-induced mechanical hypersensitivity was completely abrogated (Fig. 7d). In the CCI model, a single injection of BDT001 (5 mg/kg i.p.) reversed mechanical hypersensitivity for 2 days (Fig. 8a). As compared to pregabalin, a standard of care for PNP, BDT001 produced longer effects when considering either paw withdrawal thresholds (Fig. 8b) or pain-related behaviors (Fig. 8c). Repeated injections of BDT001 every day during 3 days fully reversed mechanical allodynia as long as the treatment continued (Fig. 8d). In agreement with the data on $Flt3^{KO}$ mice, BDT001 did not change CFA-induced inflammatory mechanical pain hypersensitivity (Fig. 8e). Neither sensory–motor functions in naive mice (Fig. 8f) nor body weight in nerve-injured mice (Fig. 8g) were affected by single and repeated BDT001 (5 mg/kg i.p.) injections during 4 days, respectively. In a preliminary pharmacokinetics study, the BDT001 plasma level reached 0.19 ± 0.06 μM ($N = 3$), 30 min after administration at a dose of 5 mg/kg i.p.

Tested at a concentration of 10 μM, BDT001 shows a high selectivity for the FLT3 kinase as neither binding to 25 diverse RTKs (Supplementary Table 3) nor inhibitory functional effects on 49 intracellular kinases (Supplementary Table 4) could be demonstrated, with the exception of Interleukin-1 receptor-associated kinase 4 (IRAK4), for which a moderate binding (64% inhibition of staurosporine binding at 10 μM) was detected. Apart from a poor metabolic stability (Supplementary Fig. 9b), BDT001 exhibits promising properties suggesting that the clinical development of close analogs is indeed feasible.

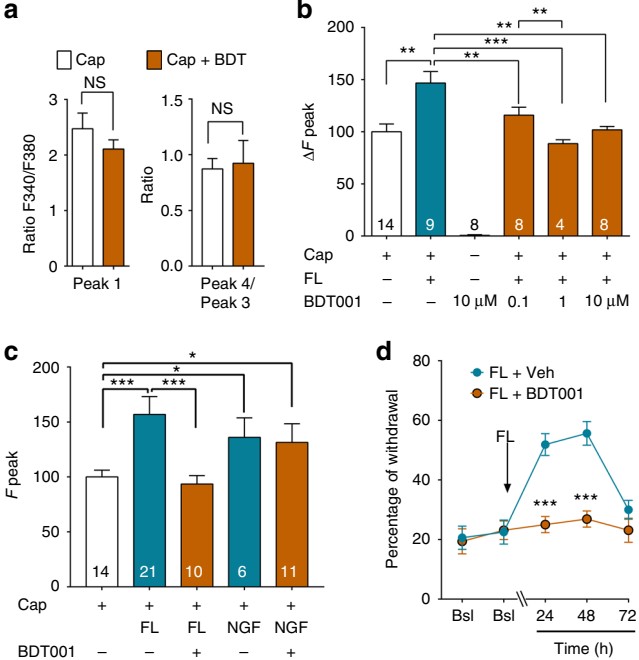

**Fig. 7** BDT001 inhibited FL-induced TRPV1 potentiation in DRG neurons in vitro and FL- induced pain symptoms in vivo. **a** In the absence of FL, BDT001 (10 μM), did not inhibit capsaicin (Cap)-induced increases in $[Ca^{2+}]_i$ levels, in a protocol identical to that described in Fig. 2a. Results are means ± s.e.m. of data from 9 neurons. Unpaired Student's $t$-test. **b** BDT001 inhibited potentiation by FL (0.056 nM) of capsaicin-induced increases in $[Ca^{2+}]_i$ levels in a concentration-dependent manner, but had no effect alone on $[Ca^{2+}]_i$ levels. **c** BDT001 (10 μM), inhibited potentiation by FL (0.056 nM), but not NGF (10 nM) of capsaicin-induced increases in $[Ca^{2+}]_i$ levels. In **b**, **c**, results are means ± s.e.m. of data obtained from the number of neurons indicated in columns and are expressed as response amplitudes normalized to capsaicin (0.5 μM) alone (Δ$F$ peak). Unpaired Student's $t$-test, *$P < 0.05$; **$P < 0.01$; ***$P < 0.001$. **d** BDT001 (5 mg/kg i.p.) inhibited mechanical pain hypersensitivity induced by FL (50 ng/10 μl, injected intrathecally). Bsl basal scores before FL injection. Results are means ± s.e.m. of data from 8 animals per group. One-way ANOVA with repeated measures and Dunnett's test, *$P < 0.05$; **$P < 0.01$; ***$P < 0.001$ vs. Bsl

## Discussion

Using different approaches aimed at selectively activating or inhibiting neuronal FLT3 signaling, we disclose a novel role for neuronal FLT3 as a master hub protein mediating neuro-immune interactions known to be crucial in PNP physiopathology[13]. Indeed, disruption of the blood–nerve barrier produced by nerve injury seems to be responsible for the presence of CD45 positive FL-expressing hematopoietic cells at the site of nerve lesion. We demonstrate that sensory neuron FLT3 is necessary and sufficient for the development and maintenance of the PNP state. Activation of FLT3 by exogenous FL in mice mimics many aspects of nerve injury-induced PNP, i.e., mechanical and thermal hypersensitivity, pain-related behaviors, rapid upregulation and potentiation of TRP family transducers in the DRG (likely involving phosphorylation[7]) that may lead to neuron hyperexcitability, and delayed transcriptional modifications in the DSC that have been reported to be associated with central pain sensitization. Disruption of FLT3 functioning by various means, e.g., gene deletion, small interfering RNAs, an AAV9 virus vector expressing *Flt3* shRNA into neurons or a newly designed inhibitor, reverse changes associated with PNP, without affecting normal sensory–motor system functioning. Our results also show

that FLT3 signaling modulates the expression of the cytokine CSF1, known to activate microglial cells in the DSC[19], a key process in central sensitization thought to underlie PNP maintenance. Hence, FLT3 in sensory neurons appears as a key upstream trigger and controller of PNP.

FLT3 activation by FL indeed requires peripheral nerve lesion, rupture of the blood–nerve barrier and presence of FL-expressing cells. Thus, our data demonstrate that FLT3 is involved in nerve injury-induced hyperalgesia and pain sensitization, but not in normal nociception nor in inflammatory pain, which does not involve nerve lesion. We have therefore discovered a specific therapeutic target for PNP.

Many high-affinity inhibitors targeting the intracellular ATP-binding site of the FLT3 kinase domain have been developed and are currently under clinical evaluation for the treatment of FLT3-mutated acute myeloid leukemia[44,45]. However, even the most potent and selective FLT3 inhibitor to date (quizartinib) still inhibits several other receptor tyrosine kinases (e.g., c-KIT, RET, PDGFRB, CSF1R) at single digit nanomolar concentrations[46]. Severe side effects associated with the therapeutic use of existing FLT3 inhibitors are acceptable in oncology patients, but not in PNP patients in the perspective of a long-lasting treatment. Using a structure-based virtual screening approach, we targeted the unique extracellular FL–FLT3 interface. This strategy of extracellular RTK inhibition[47] by low-molecular weight compounds was shown to be successful in at least two examples in which potent TrkB[48] and FGFR[49] extracellular negative allosteric modulators were discovered. We have identified a low-molecular weight compound (BDT001) as an FLT3 negative allosteric modulator that specifically inhibits FL binding and blocks FL-induced FLT3 receptor phosphorylation with moderate potency (ca. 10–20 μM) in artificial in vitro assays. However, BDT001 exhibits a much higher potency for inhibiting FL-induced effects in DRG neurons, e.g., TRPV1 potentiation compared to inhibition of binding and phosphorylation in RS4–11 cells. The concentrations of BDT001 needed to inhibit TRPV1 potentiation (0.1 μM) are similar to those reached in plasma after systemic administration at a dose that also produces robust anti-hyperalgesic and anti-allodynic effects in an PNP model in mice. At this low concentration, BDT001 does not affect any kinase binding (23 receptor tyrosine kinases tested) nor functional activity (49 kinases tested) and therefore seems to display the needed pharmacological selectivity. Indeed, BDT001 completely inhibits FL-induced mechanical hypersensitivity, an effect shown here to be dependent on FLT3 activation. This may suggest that BDT001 is a potent agent against PNP, due to the particular environment of neuronal FLT3 not present in the myeloid cell line RS4–11, or to the high sensitivity of PNP mechanisms to FLT3 inhibition. Thus, although the precise molecular mode of interaction of BDT001 with FLT3 remains to be elucidated, this compound represents a prototypical selective FLT3 inhibitor. Remarkably, BDT001 has anti-hyperalgesic effects without altering sensory–motor functions, such as nociceptive sensitivity or motor balance and coordination, which supports the selectivity of the compound. Furthermore, BDT001 does not affect inflammatory pain hypersensitivity, which suggests that BDT001 or an analog may be a specific treatment for PNP. Moreover, the effects of BDT001 last 48 h after a single administration. Such a long-lasting effect, in spite of rapid decrease in plasma drug levels, may be due to the compound remaining in a somatosensory compartment or locked in an FLT3 inactive conformation after its elimination from the systemic circulation. Another more likely hypothesis is that this long-lasting effect, may be due to prolonged inhibition of FLT3 phosphorylation, as was shown in the case of in vivo administration of the small-molecule FLT3 inhibitor sunitinib in a tumor xenograft model[50].

By specifically targeting the upstream trigger and controller of PNP, the novel FLT3-based therapy may be more efficacious than

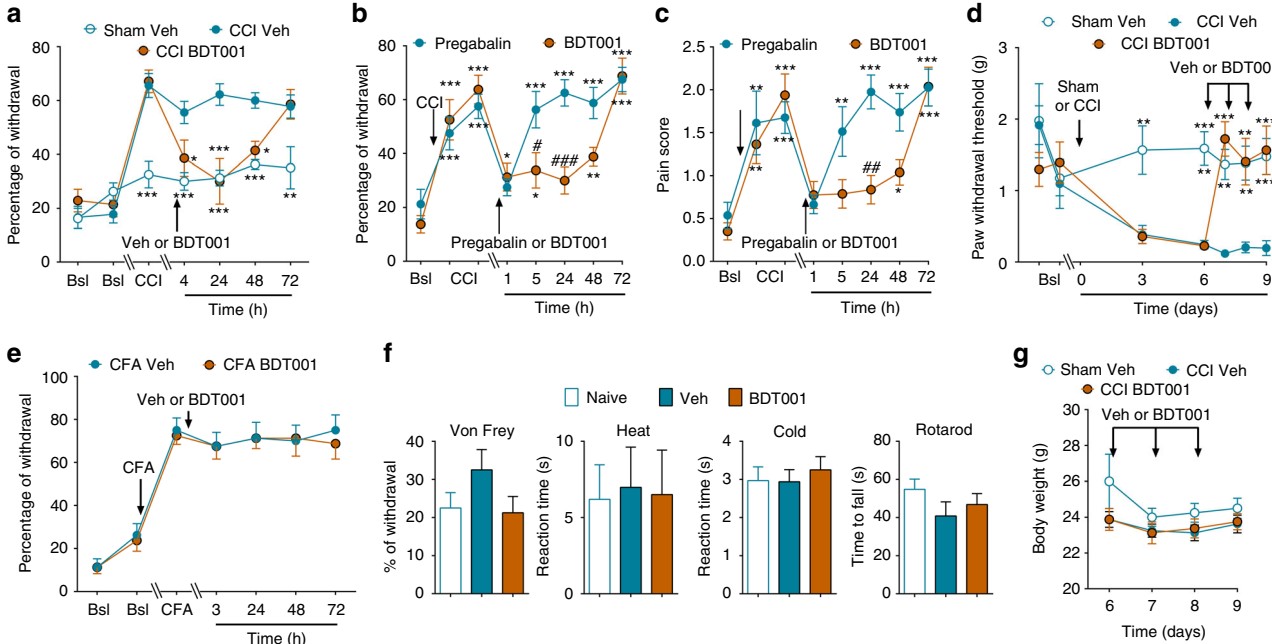

**Fig. 8** BDT001 inhibited CCI-induced pain symptoms in vivo, without altering body weight in CCI-mice or sensory–motor behaviors in non-injured mice (**a**) BDT001 (5 mg/kg i.p.) inhibited mechanical hypersensitivity in CCI-mice. After two basal scorings (Bsl), CCI or Sham was performed and mechanical hypersensitivity measured 10 days later (CCI). Eleven days after sham or CCI, vehicle (Veh) or BDT001 (5 mg/kg i.p.) was injected and mechanical hypersensitivity measured 4, 24, 48, or 72 h after injection. **b**, **c** Comparison of BDT001 (5 mg/kg i.p.) and pregabalin (10 mg/kg) on mechanical hypersensitivity (**b**) and pain-related behaviors (**c**) in CCI-mice. **a–c** One-way ANOVA with repeated measures and Dunnett's test, *$P < 0.05$; **$P < 0.01$; ***$P < 0.001$ vs. Bsl; two-way ANOVA with repeated measures and Bonferroni's test #$P < 0.05$; ##$P < 0.01$; ###$P < 0.001$ vs. FL-Veh or CCI-Veh or pregabalin. **d** BDT001 (5 mg/kg i.p.) inhibited punctuate tactile pain in CCI-mice. Treatment with BDT001 or vehicle (Veh) was administered immediately after scoring 6, 7, and 8 days after CCI. Two-way ANOVA with repeated measures and Bonferroni's test. *$P < 0.05$; **$P < 0.01$; ***$P < 0.001$ vs. CCI Veh. **e** BDT001 (5 mg/kg i.p.) did not affect mechanical hypersensitivity to chronic inflammatory pain induced by paw injection of complete Freund's adjuvant (CFA). Two-way ANOVA with repeated measures (treatment effect, $p = 0.8265$). **f** In non-injured mice, BDT001 (5 mg/kg i.p.) did not significantly change mechanical sensitivity and heat and cold sensitivity, as measured by the von Frey, Hargreaves and acetone tests, respectively, nor performance in the rotarod test. One-way ANOVA ($p = 0.1764$, $p = 0.3478$, $p = 0.7772$, and $p = 0.2908$, respectively). **g** BDT001 did not significantly change body weight in mice utilized in **d**, One-way ANOVA with repeated measures ($p = 0.9285$). **a–g** Results are means ± s.e.m. of data from 8 animals per group

current therapies targeting either non-specific PNP mechanisms (e.g., anti-epileptics or antidepressants), or mechanisms operating on narrow aspects of PNP, such as excitability of sensory neurons (e.g., voltage-dependent sodium channel blockers) or inflammation (e.g., interleukins or chemokines inhibitors).

In conclusion, our data support peripheral sensory neuron FLT3 as an innovative and specific target for PNP management. BDT001, a rationally designed FLT3 negative allosteric modulator, is consequently a novel prototypical therapeutic tool to alleviate PNP symptoms.

## Methods

**Animals**. Experiments were performed in C57BL/6 naive mice (Janvier, France) or mice carrying a homozygous deletion of *Flt3* (*Flt3*KO mice)[24] and their littermates (WT) weighing 25–30 g. All the procedures were approved by the French Ministry of Research (authorization #1006). Animals were maintained in a climate-controlled room on a 12 h light/dark cycle and allowed access to food and water ad libitum. Male and female mice were first considered separately in behavioral procedures. Both sexes showed mechanical hypersensitivity of same intensity after intrathecal FL injection and nerve injury and were similarly affected by *Flt3* deletion (ANOVA followed by Bonferroni's test, n = 8 for both sexes and genotypes for each experiment). Thereafter, experiments were performed only on male mice.

**Chronic pain models**. Four different models of peripheral neuropathic pain and one model of chronic inflammatory pain were used. All surgical procedures were performed under deep isoflurane anesthesia.

The CCI model was performed as described previously[29] and adapted for mice[51]. Briefly, skin was incised and the sciatic nerve was exposed unilaterally at the mild-high level by dissecting through the biceps femoris. Three ligations (catgut 6.0) were loosely tied around the sciatic nerve with about 1 mm spacing to reduce

blood flow. The skin was then closed with staples. In sham-operated animals the sciatic nerve was exposed without ligation.

The SNI procedure was performed as already described[41]. Briefly, the three terminal branches of the sciatic nerve were exposed after incision of the lateral surface of the thigh through the biceps femoris muscle. Then, the common peroneal and the tibial nerves were tightly ligated with 6.0 silk thread and sectioned distal to the ligation. The sural nerve was left intact.

The SNL procedure was performed as described previously[42]. Briefly, the L6 transverse process was removed to expose the L4 and L5 spinal nerves. The L5 spinal nerve was then isolated and tightly ligated with 6.0 silk thread. For sham operations, the L5 spinal nerve was exposed but not ligated.

The induction of oxaliplatin-induced peripheral neuropathy was performed according to Descoeur et al.[52] Oxaliplatin (3 mg/kg) was dissolved in saline and intraperitoneally injected in the animals. Sham animals were injected with an equivalent volume of saline.

The model of complete Freund adjuvant (CFA)-induced pain[53] has been used for assessing chronic inflammatory pain. Briefly, under isoflurane anesthesia, an intraplantar injection (20 µl) of a solution of 1 mg of mycobacterium tuberculosis (Sigma-Aldrich) per ml was performed in the left hind-paw of wild-type or *Flt3*KO animals.

**Behavioral testing**. Before testing, mice were acclimatized for 60 min in the temperature and light-controlled testing room within a plastic cylinder or on wire mesh. Experimenters were blinded to the genotype or the drug administered.

Three different types of tests were performed to evaluate mechanical sensitivity.

A 0.6 g-von Frey filament was used to test hind-paw mechanical hypersensitivity. Sharp withdrawal of the stimulated hind-paw was considered as a positive response. The procedure was applied 10 times and the percentage of positive responses was calculated. According to Ducourneau et al.[54], we quantified pain-related behaviors in response to mechanical stimulation by recording a pain score as followed. 0; no withdrawal, 1; movements of the toes without withdrawal, 2; slow withdrawal, 3; sharp withdrawal, 4; withdrawal with nociceptive behaviors such as flinching, shaking/or licking. Tactile withdrawal threshold was also determined in response to probing of the hind-paw with eight calibrated von Frey

filaments (Stoeling, Wood Dale, IL, USA) in logarithmically spaced increments ranging from 0.41 to 15 g (4–150 mN). Filaments were applied perpendicularly to the plantar surface of the paw. The 50% paw withdrawal threshold was determined in grams by the Dixon nonparametric test[55]. The protocol was repeated until three changes in behavior occurred. In the dynamic von Frey procedure, animals were placed on a metal mesh surface in an enclosed area. A stainless steel filament (0.5 mm diameter) was automatically applied to the hind-paw. The filament exerted an increasing force to the plantar surface until paw withdrawal. The latency until withdrawal, in seconds, and the force, in grams, at which the paw was withdrawn were recorded.

For assessing cold sensitivity, acetone (60 µl) was applied first on the left hind-paw for all animals then on the right hind-paw. The time spent licking or biting the paw was recorded with a stopwatch and reported as the cumulative time of licking/biting for the two hindpaws. A cut-off time of 45 s was used in each trial.

For assessing heat sensitivity, a radiant heat source (plantar test Apparatus, IITC Life Science, Woodland Hills, USA) was focused onto the plantar surface of the paw. The paw withdrawal latency was recorded and nociceptive behaviors were scored as followed: 1: no reaction, 2; paw withdrawal, 3; sharp withdrawal with shaking or licking of the paw. Each paw was tested 3 times with 10 min-intervals between each trial. A maximal cut-off of 20 s was used to prevent tissue damage. The data are expressed as paw withdrawal latency or pain score calculated by dividing the paw withdrawal latency by the score of nociceptive behavior measured for each animal.

Intraplantar capsaicin (10 µl of a solution at 5 µM in 1%-DMSO) was performed in unanaesthetized mice. Immediately after the injection, each animal was observed for 20 min and spontaneous pain-related behaviors were evaluated through the time spent in shaking and licking the injected hind-paw. Data are represented as the duration of the responses. After a first injection of capsaicin associated with intrathecal saline or FL (500 ng/10 µl), capsaicin injection was repeated three times, 30 min, 3 h, and 24 h after the first injection.

The model of formalin-induced pain[56,57] was used to assess acute inflammatory pain as followed. In wild-type or $Flt3^{KO}$ unanaesthetized mouse, an intraplantar injection (10 µl) of 2.5% formalin was performed in the left hind-paw. Spontaneous nociceptive behaviors were evaluated for 45 min by measuring the time spent in shaking and licking the injected hind-paw. Data are expressed as duration of responses every 5 min.

In the Rotarod test, the speed was set at 10 rpm for 60 s and subsequently accelerated to 80 rpm over 5 min. The time taken for mice to fall after the beginning of the acceleration was recorded.

Conditioned place preference (CPP) was performed as follows. Tonic-aversive state in neuropathic pain can be unmasked by the administration of non-rewarding and rapidly-acting analgesic drugs such as clonidine. All experiments were conducted by using the single trial CPP protocol as described previously for rodents[40,58]. CPP apparatus (Bioseb, Vitrolles, France) consists of two equally sized chambers (20 × 18 × 25 cm) interconnected by a rectangular corridor (20 × 7 × 25 cm). The chambers are differentiated by the wall pattern (dotes versus stripes) and color (different shades of gray versus black). Fourteen days after CCI surgery, animals went through a 3-day pre-conditioning period with full access to all chambers for 20 min. On day 3, a pre-conditioning bias test was performed to determine whether a preexisting chamber bias existed. In this test, mice were placed into the middle chamber and allowed to explore open field with access to all chambers for 15 min. No animal spending more than 80% or <20% of the total time in an end chamber was found and all animals tested were retained for this CPP experiment. On conditioning day (day 4), mice first received intrathecal saline (10 µl) paired with a randomly chosen chamber in the morning. Four hours later, the same animals received intrathecal clonidine (10 µg/10 µl.) paired with the other chamber in the afternoon. Conditioning sessions lasted 15 min, each mice did not have access to the other chamber. On the test day (d5), 20 h after the afternoon pairing, mice were placed in the middle chamber of the CPP box with all doors open so animals could have free access to all chambers. The time spent in each chamber was recorded for 20 min for analysis of chamber preference.

**Drug delivery**. BDT001 (5 mg/kg, i.p.) or pregabalin (10 mg/kg, i.p., R&D Systems Europe, France) was administered intraperitoneally. For continuous siRNA intrathecal delivery, a polyurethane catheter (Alzet #0007743) with the following specifications: 2.5 cm 32 G (0.23 mm OD; 0.09 mm ID) connected to 1 cm (0.76 mm OD; 0.38 mm ID), connected to a 2.5 cm (1.02 mm OD; 0.61 mm ID) ALZET connection with Teflon-coated stylet wire was inserted into the subarachnoid space between S1 and L6 level and anchored with histoacryl tissue adhesive to L6 vertebrae and attached to muscles surrounded the spine with 4.0 silk suture. After a 7 day-recovery period, the minipump was connected to the catheter and implanted into the dorsal subcutaneous space. Behavioral experiments were performed 10 days after minipump (Model 1002, Alzet Osmotic pump, Charles River) implantation. For FL or virus acute intrathecal injection, a 30 G needle attached to a microsyringe was inserted between L4 and L5 vertebrae in lightly restrained, unanaesthetized mice. The reflexive tail flick was used to confirm the punction. A total volume of 10 µl was injected. After intrathecal injection, 2 mice were excluded from the data analysis because they presented motor dysfunction and/or paralysis. In Fig. 2g, FL injection was performed in the paw. For intra-nerve injection (Supplementary Fig. 3a), under isoflurane anesthesia, the left sciatic nerve was exposed as described above for CCI surgery. A Hamilton syringe (10 µl) was connected to an electronic pump by polyethylene tubing. To avoid damaging the

nerve we used a 33 G needle to disrupt the epineurium of the sciatic nerve and a polyurethane tubing (Alzet #0007743) was inserted to slowly deliver the solutions into the sciatic nerve. One µl of a solution of FL (50 µg/ml) or saline was injected in mice at a rate of 30 µl/h. After the injection, the skin was closed with staples. After intrathecal injection, no mouse presented motor dysfunction and all the animals were retained for the data analysis. In Supplementary Fig. 3b, FL (5 mg/100 µl) was injected intravenously.

**FLT3 transcript knockdown experiments with siRNAs**. Scrambled control small interfering (siRNA) based on the Flt3 sequence and a pool of 4 specific siRNAs against $Flt3$ ($Flt3$-siRNA) were used (On-target Plus SMART pools from Dharmacon, Perbio Science, Brebières, France; Dharmacon Catalog L-002000-00-0005 targeting FLT3). Anti-$Flt3$ siRNA was validated in vitro in HEK293M cells transfected with a SNAP-tagged-$Flt3$ cDNA and the relevant siRNA or a control siRNA. HEK293M cells were maintained in DMEM Glutamax (Invitrogen) supplemented with antibiotics (penicillin 50 U/ml, streptomycin 50 µg/ml) and 10% heat-inactivated Fetal Calf Serum. Transfection mixes were prepared using 0.5 µg of mouse SNAP-FLT3 plasmid, 9.5 pmol of mouse $Flt3$-siRNA, 5 µl of Lipofectamine 2000 (Invitrogen) and 1000 µl Opti-MEM per well. The SNAP fluorescent signal was detected after incubation with Tb-labeled benzylguanine (100 nM), using an advanced fluorescence microplate reader (CLARIOstar, BMG Labtech) equipped with a HTRF optic module allowing a donor excitation at 337 nm and a signal collection both at 620 nm. A frequency of 300 flashes/well was selected for the lamp excitation. For in vivo injections, 5 µg of specific or non-specific siRNAs were complexed with 1.8 µl of 200 µM linear low-molecular weight polyethylenimine ExGen 500 (Euromedex, Souffel-weyersheim, France). $Flt3$-siRNA (12.53 ng/ml) or a scrambled siRNA was administered via osmotic minipump infusion (0.25 µl/h) for 4 days.

**Real-time PCR**. Mice lumbar (L4–L6) dorsal root ganglia and dorsal spinal cord were dissected at different stages post-surgery and stored at −80 °C until RNA was extracted using the RNAqueous-4PCR Kit (Ambion). One µg of total RNA was reverse-transcribed with 100 U of Superscript II reverse transcriptase (Invitrogen), 5 µM random hexamers (Promega), 0.5 mM of each dNTPs (Promega), 10 mM of dithiothreitol, and 20 U of recombinant RNase inhibitor (Promega) 1 h at 37 °C. Real-time PCR was carried out as described previously[59] using SYBR Green I dye detection on the Light Cycler system (Roche Molecular Biochemicals). PCR reactions were carried out in 96-well plates in a 10 µl volume containing 3 µl of cDNA product (final dilution 1/30), 0.5 µl of forward and reverse primers, and 2 µl of QuantiTect SYBR Green PCR Master Mix (Roche Diagnosis). Amplified products were sequenced at least once (Beckman Coulter Genomics, UK). The relative amounts of specifically amplified cDNAs calculated on at least three independent experimental replicates using the delta-CT method[59–61] were normalized with RNA polymerase II polypeptide J (Polr2j) and DEAD box polypeptide 48 (Ddx48) as stable control genes.

Sequences of the primer pairs used are as follows:

Polr2j: F-ACCACACTCTGGGGAACATC, R-CTCGCTGATGAGGTCT GTGA; NM_011293

Ddx48: F-GGAGTTAGCGGTGCAGATTC, R-AGCATCTTGATAGCCC GTGT; NM_138669

Atf3: F-ACAACAGACCCCTGGAGATG, R-CCTTCAGCTCAGCATTCACA; NM_007498

TrpV1: F-GGATCCCTCGGAAGAAGAAG, R-GCAGGACAAGTGGGACA GAT; NM_001001445

TrpA1: F-GCGGAGACTTGGACATGATT, R-TCTGTGAAGCAGGGTCTC CT; NM_177781

NpY: F-TGGACTGACCCTCGCTCTAT, R-TGTCTCAGGGCTGGATCTCT; NM_023456

Iba1 alias Aif1: F-GGATCAACAAGCAATTCCTCGA, R- AGCCACTGGA CACCTCTCTA; NM_019467

Gfap: F-GCCACCAGTAACATGCAAGA, R- GCTCTAGGGACTCGTTC GTG; NM_010277

Cd11b alias Itgam: F-ACATGTGAGCCCCATAAAGC, R-AATGACCC CTGCTCTGTCTG; NM_001082960

Csf1: F-GCCGCTGTGTCCGAACTTTCCA, R-GATCCCTCATGCTGCTCCAC; NM_001113529

Csf1R: F-TCTTGTGTGGCCAGCAATGA, R-GCTTGCGCTGGTCTTC AAAG; NM_001037859

**Adult sensory neuron culture**. For both whole-cell patch-clamp recordings and calcium imaging, neuron cultures were established from lumbar (L4–L6) dorsal root ganglia[59]. Briefly, ganglia were successively treated by two incubations with collagenase A (1 mg/ml, Roche Diagnostic, France) for 45 min each (37 °C) and trypsin-EDTA (0.25%, Sigma, St Quentin Fallavier, France) for 30 min. They were mechanically dissociated through the tip of a fire-polished Pasteur pipette in neurobasal (Life Technologies, Cergy-Pontoise, France) culture medium supplemented with 10% fetal bovine serum and DNase (50 U/ml, Sigma). Isolated cells were collected by centrifugation and suspended in neurobasal culture medium supplemented with 2% B27 (Life Technologies), 2 mM glutamine, penicillin/streptomycin (20 U/ml, 0.2 mg/ml) plated at a density of 2500 neurons per coverslip and were

incubated in a humidified 95% air-5% $CO_2$ atmosphere at 37 °C. We used 6 and 21 animals for path-clamp and calcium imaging experiments, respectively.

**Whole-cell patch-clamp recordings from isolated DRG neurons.** Whole-cell voltage-clamp was performed on small and medium somatic diameter (15–25 μm) DRG neurons after one day in vitro (1 DIV) at 37 °C. Currents were recorded with an RK-400 amplifier (Biologic) and computed with a Digidata 1322 A analogue interface (Axon Instruments) and the pClamp software (Clampex 8.02; Axon Instruments). Signals were filtered at 3 and sampled at 5 kHz. Series resistances were corrected to 60–80%. The membrane voltage was held at −60 mV. Glass electrodes (2.5–3.5 MΩ) were made from capillary glass, using a Narishige puller, and coated with paraffin wax to minimize pipette capacitance. The Standard External Solution (SES) contained: 140 mM NaCl, 5 mM KCl, 2 mM $CaCl_2$, 2 mM $MgCl_2$, 10 mM HEPES, 10 mM glucose (pH adjusted to 7.4 with NaOH and osmolarity between 300 and 310 mOsm). Recording pipettes were filled with the following solution: 140 mM KCl, 2 mM MgCl2, 5 mM EGTA, 10 mM HEPES (pH adjusted to 7.4 with KOH and osmolarity between 300 and 310 mOsm). Capsaicin (500 nM-1 μM) and FL (100 nM) were dissolved in extracellular solution. Sequencial applications were: SES (1 min)—capsaicin 1 (15 s)—SES (3 min)—capsaicin 2 ± FL (15 s)—SES (3 min)—capsaicin 3 ± FL (15 s)—SES (1 min).

**Calcium imaging.** For calcium imaging video microscopy $[Ca^{2+}]_i$ fluorescence imaging, DRG neurons were loaded with fluorescent dye 2.5 μM Fura-2 AM (Invitrogen, Carlsbad, CA) for 30 min at 37 °C in standard external solution contained: 145 mM NaCl, 5 mM KCl, 2 mM $CaCl_2$, 2 mM $MgCl_2$, 10 mM HEPES, 10 mM glucose (pH adjusted to 7.4 with NaOH and osmolarity between 300 and 310 mOsm). The coverslips were placed on a stage of Zeiss Axiovert 200 inverted microscope (Zeiss, München). Observations were made at room temperature (20–23 °C) with a ×20 UApo/340 objective. Fluorescence intensity at 505 nm with excitation at 340 and 380 nm were captured as digital images (sampling rates of 0.1–2 s). Regions of interest were identified within the soma from which quantitative measurements were made by re-analysis of stored image sequences using MetaFluor Ratio Imaging software. $[Ca^{2+}]_i$ was determined by ratiometric method of Fura-2 fluorescence from calibration of series of buffered $Ca^{2+}$ standards. Neurons were distinguished from non-neuronal cells by applying 25 mM KCl, which induced a rapid increase of $[Ca^{2+}]_i$ only in neurons. Pulses of capsaicin were applied at 2 min-intervals and capsaicin and drugs added after the third pulse. The same temporal stimulation protocol (Fig. 2a) was used for TRPV1 (capsaicin), TRPA1 (cinnamaldehyde), and TRPM8 (menthol). To allow visualization of cells transfected with siRNA (see above for the preparation of siRNA), 3 mM dextran-tetramethyl rhodamine (Invitrogen, Cergy-Pontoise, France) was added to the 5% glucose solution containing the RNA–polymer complex.

For data analysis, amplitudes of $[Ca^{2+}]_i$ increases, $\Delta F/F_{max}$, caused by stimulation of neurons with capsaicin were measured by subtracting the "baseline" $F/F_{max}$ (mean for 30 s prior to capsaicin addition) from the peak F/Fmax achieved on exposure to capsaicin. In the absence of any treatment the distribution of these ratios was well fitted by a normal distribution.

**In situ hybridization.** Sense and antisense digoxigenin (DIG)-labeled RNA probes were generated from a mouse Flt3 cDNA clone (IRAMp995N1310Q, Genome-CUBE Source Bioscience) in a 20 μl reaction containing 1 μg of linearized plasmid (digested with HindIII and NheI, respectively) using the DIG RNA labelling mix (Roche Diagnostics) and T7 and Sp6 RNA polymerases respectively (Promega) following the manufacturer's instructions. DIG-labeled RNA probes were purified on MicroSpin G50 columns (GE Healthcare). Naive or CCI-injured WT and $Flt3^{KO}$ mice were euthanized by $CO_2$ inhalation followed by cervical dislocation. L4–L6 lumbar dorsal root ganglia and spinal cord were dissected in phosphate-buffered saline (PBS), and fixed for 1 and 3 h, respectively, in 4% paraformaldehyde (PFA) at room temperature. Tissues were rinsed twice in PBS before immersion overnight in 25% sucrose/PBS at 4 °C. In situ hybridization was performed with standard procedures on transverse sections of 14 μm as described[62]. Image acquisition was done using a Hamamatsu NanoZoomer using NDP view software.

**Virus transfection.** The following oligonucleotides were used to build the AAV-GFP-shRNA plasmids: shFLT3 (GATCGGTGTCGAGCAGTACTCTAAATCAA-GAGTTTAGAGTACTGCTCGACACCTTTTT (top)); AGCTAAAAAGGTGTC-GAGCAGTACTCTAAACTCTTGATTTAGAGTACTGCTCGACACC (bottom); Sigma-Aldrich (TRC00000378670); shcontrol (GATCCAACAAGATGAAGAGC ACCAATCAAGAGTTGGTGCTCTTCATCTTGTTGTTTTTT (top)); AGCTAA AAACAACAAGATGAAGAGCACCAACTCTTGATTGGTGCTCTTCATCTT GTTG (bottom).

The oligonucleotides were annealed and cloned BamHI–HindIII in a vector containing the U6 promoter sequence and the hGH polyadenylation sequence upstream and downstream, respectively, of the BamHI site. The U6-shRNA-hGH sequences were then cut using PmlI and HpaI and cloned in the pAAV-CMV-GFP vector (Cell Biolabs Inc.) linearized with PmlI. All the constructs were checked by DNA sequencing. The viruses were produced by the Viral Vector Production Unit of Barcelona, Spain. For in vivo experiments, 10 μl of AAV-GFP-shFLT3 or AAV-GFP-sh control solution were intrathecally injected in lightly restrained,

unanaesthetized mice, as described above. The titer of the virus solution was $5.8 \times 10^{12}$ genome copies (gc)/ml.

**Immunohistochemistry.** Mice were transcardially perfused with PBS followed by 4% formaldehyde for all experiments except for the AVV9-GFP ShFLT3 experiment where mice were perfused with 1% saline buffer. DRG (L4-L6), spinal cord (lumbar segment), and sciatic nerves were collected and post-fixed in 4% paraformaldehyde between 10 min and 2 h depending on the antibody and tissue before being cryo-protected in 30% sucrose in PBS. Tissues were then frozen in O.C.T (Sakura Finetek). Sections (12 μm) were prepared using the Cryostat Leica CM2800E. For immunostaining, frozen sections were blocked and permeabilized in $Ca^{2+}/Mg^{2+}$-free PBS (PBS 1× −/−) containing 10% donkey serum and 0.1% or 0.3% Triton x-100 during 30 min. The sections were then incubated with primary antibodies, at 4 °C overnight in PBS 1× −/− containing 1% donkey serum and 0.1%, 0.01, or 0.03% Triton x-100. After extensive wash in PBS 1× −/− (3 times for 10 min minimum each) sections were incubated with appropriate secondary antibody conjugated to AlexaFluor and Hoechst (Sigma 1 μg/ml), in the same buffer as the primary antibodies, at room temperature for 1 h and then washing (three times for 15 min each) before mounting with Mowiol. Images were acquired under a Zeiss LSM 510 and 700 Confocal microscope with ZEN software (Carl Zeiss Microscopy).

**Antibodies.** The primary antibodies that were used in the study are listed below, together with manufacturer, catalog number, dilutions used and references to external publications using the antibodies. Each antibody was validated in the specific condition and tissue applied in the study by routine controls (omission of primary or secondary antibody) and by checking that the pattern of immunolabelling distribution or the number of identified neurons in DRG and DSC corresponds to the published data, for which an abundant literature exists. For instance, it was checked that immunoreactivity against commonly used glial markers (GFAP, IBA1) and neurons (TUJ1, NeuN, NPY, CSF1) is restricted to glial cells and neurons, respectively, as identified by morphometric features. Most of these antibodies have been previously used in our lab conditions[62–65]. In addition, the anti-FLT3 antibody, which has not been extensively used, was validated by using $Flt3^{KO}$ mice (Supplementary Figure 3c).

The following primary antibodies were used: anti-TUJ1 (Sigma T9026 or T2220, mouse or rabbit 1:1000), anti-ATF3 (Santa Cruz sc-188, rabbit 1:500), anti-NPY (Cell Signaling 11976S, rabbit 1:500), anti-GFAP (Dako Z0334, rabbit 1:500 in spinal cord or 1:1000 in DRGs), anti-IBA1 (Wako 019-19741, rabbit 1:500), anti-CSF1 (R&D System AF416, goat 1:500), anti-CD45 (Millipore 05-1416, rat 1:100), anti-CD11b (DSHB M1/70.15.11.5.2, rat 1:100), anti-CD3 (BD Pharmagen 555273, rat 1:500), anti-CD68 (Serotec MCA1957GA, rat 1:100), anti-FLT3 ligand (Bioss 5905R, rabbit 1:100 or Cell Signaling Technology, clone 8F2 3462, rabbit 1:200), anti-GFP (Invitrogen A6455, rabbit 1:2000 orAbcam Ab13970, chicken 1:2000), anti-S100β (Sigma S2532, mouse 1:1000), anti-NF200 (Sigma N4142, rabbit 1:1000), anti-cRet (R&D System AF482, goat 1:20 unmasking citrate buffer, see refs. [26,65,66]), anti-TrkC (R&D System AF1404, goat 1:1000), anti-TrkB R&D System AF1494 (goat 1:1000), anti-TrkA (Millipore 06574, rabbit 1:500), anti-TRPV1 (Sigma V2764, rabbit 1:1000). To identify IB4-binding cells, biotynylated IB4 (Sigma 2140, 1:100), anti-NeuN (Millipore ABN90, guinea pig 1:2000) and Extravidin-conjugated FITC (Sigma E2761, 1:400) were used in place of primary and secondary antibodies. The following secondary antibodies were used anti-rabbit AlexaFluor 594/488/647 (goat or donkey 1:1000), anti-rat AlexaFluor 594/488 (donkey 1:1000), anti-goat AlexaFluor 594 (donkey 1:1000), anti-guinea pig AlexaFluor 647 (donkey 1:1000) and anti-mouse AlexaFluor 594/647 (donkey 1:1000), anti-chicken AlexaFluor 488 (goat 1:1000) from Life Technologies, and anti-guinea pig AlexaFluor 647, donkey 1:1000, from Jackson Immuno Research.

**Immunostaining quantification.** The number of neurons expressing the various molecular markers of sensory neurons (subtypes) was determined by counting cells with neuronal morphology or in some experiments with TUJ1 immunostaining, e.g., Fig. 4d. A minimum of six sections from lumbar DRGs were counted from at least three animals for each genotype or condition. For the GFAP immunostaining in DRG the same method was used, but a positive signal was recorded when a neuronal cell body was encircled at least two-thirds by GFAP staining. The total number of neuronal cells in the DRG were counted to determine a percentage of GFAP-encircled neuron cell bodies[63].

To quantify GFAP and IBA1 immunostaining in the dorsal horn of the spinal cord, a minimum of 10 sections from lumbar spinal cord were evaluated using Image J software. A density threshold was set above background level to identify positively stained structures in each condition or/and genotype. The area occupied by these structures was measured as positive area. An average percentage of area relative to the total area of the spinal dorsal horn of the section was obtained for each animal across the different tissue sections. Four animals have been excluded from the immunohistochemistry analysis (Fig. 3k, l) because of poor quality of slices.

**Western Blot.** Mice received an intraperitoneal injection of pentobarbital and transcardially perfused with PBS. After isolation, tissues were mechanically homogenized at 4 °C in NP40 buffer (1% NP40, 150 mM NaCl; 50 mM Tris-HCl, pH = 7.8 and protease inhibitor). Lysates were clarified for 10 min at 4°c at

12,000×*g*. After protein quantification using a BCA kit (Thermofisher, France), lysates were run on SDS-PAGE and transferred to nitrocellulose membrane. Antibodies: Anti-Flt3 was purchased from Stress Signaling and anti-tubulin from Sigma. After incubation with primary and secondary antibodies, immunodetection was performed using chemiluminescent reagent (SuperSignal West Femto-ECL, Thermofisher). Quantification were done with Image Lab software (Biorad).

**Production of human recombinant FL (rh-FLT3-L)**. Recombinant FL was produced in the *E. coli* Rosetta (DE3) strain (Novagen) in our laboratory using the pET15b-rhFL plasmid according to the protocol described[64] with some minor modifications (see Supplementary Methods, Chemicals and Reagents Materials and Methods). The rh-FL was checked for endotoxin content using the Pyrogen Recombinant Factor C endotoxin detection assay from LONZA (Walkersville MD, USA) and was found free of endotoxins.

**Data and statistical analyses**. All experiments were randomized. Data are expressed as the mean ± s.e.m. All sample sizes were chosen based on our previous studies except for animal studies for which sample size has been estimated via a power analysis using the G-power software. The power of all target values was 80% with an alpha level of 0.05 to detect a difference of 50%. Statistical significance was determined by analysis of variance (ANOVA one-way or two-way for repeated measures, over time). In all experiments in which a significant result was obtained, the *F* test was followed by Bonferroni's post-hoc test for multiple comparisons or by Dunnett's post-hoc test to compare with the control group, as appropriate. In case of two experimental groups, unpaired two-tailed *t*-test was applied. For cell counting, and area quantification, statistical analyses were performed using Mann–Whitney test. The applied statistical tests are specified in each figure legend.

**Pharmacophore determination**. A FL-FLT3 interaction pharmacophore was generated starting from the X-ray structure of the FL-FLT3 complex (PDB id 3QS7) using the 'Receptor-Ligand Pharmacophore Generation' protocol[65] in DiscoveryStudio v3.1 (Accelrys, San Diego, USA). In the first step, all potential pharmacophore features of the ligand (FL: H8-S13) were identified. Six standard pharmacophore features were considered: hydrogen-bond acceptor, hydrogen-bond donor, positive ionizable, negative ionizable, hydrophobic, and ring aromatic. In the second step, we pruned all features that did not match observed FL-FLT3 interactions using a set of prefixed topological rules[65]. Last, an exclusion sphere was added for each FLT3 residue. The size of the exclusion sphere was proportional to the number of neighboring protein atoms within a 4–5 Å distance range. The final pharmacophore was composed of four hydrogen-bond acceptor features, one hydrogen-bond donor feature and 12 exclusion spheres.

**Virtual screening**. An in-house developed Bioinfo database[66] of 2.9 million commercially available drug-like compounds (v.2011 release) was first filtered in Pipeline Pilot v.8.5 (Accelrys) to retrieve molecules with the following properties: h-bond donor count ≥1, h-bond acceptor count ≥4, number of rotatable bonds ≤10, number of aromatic rings ≥1, polar surface area ≤90Å$^2$ and predicted aqueous solubility ≥50 µM. The resulting 253,193 compounds were then converted into three-dimensional coordinates using Corina v.3.1 (Molecular Networks GmbH, Erlangen, Germany) to yield a total of 343,847 unique isomers. All compounds were ionized at physiological pH with Filter v.2 (OpenEye Scientific Software, Santa Fe, USA). A maximum of 250 conformers were generated in DiscoveryStudio for each compound using the "FAST" parameter settings with an upper energy threshold of 20 kcal/mol with respect to the global energy minimum. The "Screen Library" protocol in DiscoveryStudio was used to screen the compound library against the FLT3-FL interaction pharmacophore model using rigid fitting. A total of 285 different compounds with a fitness value ≥3.0 were selected and further clustered by maximum common substructures in MedChemStudio v3.1 (Simulation Plus, Lancaster, U.S.A). A representative compound (best fitness score) of each of the 28 clusters described by at least 3 compounds was finally selected for purchase and biological evaluation.

**Data availability**. The authors declare that all data supporting the findings of this study are available within the paper and its supplementary information files or are available from the authors upon request.

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

## Acknowledgements

We thank Dr Lemischka (The Black Family Stem Cell Institute, Icahn School of Medicine at Mount Sinai, New York) for providing the *Flt3* knockout mice, the TechMedIll platform (Illkirch) for ADMET studies, Alexandre Boissonas (Inserm U1135, Paris) for expertise in immunology and Bernard Pau (Biodol Therapeutics) for stimulating discussion on the subject. We also thank the different technical platforms of Institut des Neurosciences de Montpellier, especially the animal care facility supervised by Jérôme Sarniguet and the functional exploration platform. This work was supported by INSERM and Montpellier University and grants from La Region Languedoc-Roussillon, the European Commission, the SATT AxLR, the National Research Agency (grant ANR-15-CE18-0009-01), and Biodol Therapeutics.

## Author contributions

C.R. designed, performed, supervised, and analyzed behavioral experiments to which C. S., A.J., A.T., and H.H. contributed. Ch.S. designed, performed, and analyzed calcium imaging experiments to which J.M. contributed. I.M. designed and performed Q-PCR analyses. L.D. performed histology and analyzed the data. J.P.-L. performed in vitro experiments with siRNA and performed and analyzed BDT001 binding and phosphorylation experiments. O.L. performed and analyzed electrophysiology experiments. J.P.-P., E.T., and F.C-S designed and supervised the development of HTRF in vitro assays. S.V. performed in situ hybridization experiments. S.M. performed cell culture for calcium imaging. A.M. designed, performed, and analyzed Western blot experiments and designed shRNA for virus transfection. W.J. performed molecular biology experiments for virus transfection. M.S supervised organic synthesis done by Y.P. D.R. was in charge of the design and medicinal chemistry optimization program. C.R., I.M., A.M., P.S, P.C. D.R., and J.V. participated in the interpretation of the results and in the manuscript preparation. F.M. contributed to initial aspects of the work and participated in the final preparation of manuscript. A.P. and P.C. provided critical input on study design and interpretation. J.V. conceived the project and supervised all experiments.

## Additional information

**Competing interests:** C.S. and J.P.-L. are currently full-time employees at Biodol Therapeutics. P.S. serves as Chief Scientific Officer at Biodol Therapeutics. D.R. and J.V are inventors of patents claiming FLT3 inhibitors and their use for the treatment of neuropathic pain and are co-founders of Biodol Therapeutics. The remaining authors declare no competing interests.

