## [Peer review file · Nature Communications]

Reviewers' comments:

Reviewer #1 (Remarks to the Author):

This report describes a very extensive and impressive series of experiments designed to highlight the central role of the FL/FLT3 cytokine signaling system in the establishment and maintenance of neuropathic pain as assessed using rodent (mice) pain models. FLT3 activation in DRG neurons resulted in rapid pain due to transactivation of TRP channels as well as longer lasting effects due to regulation of gene expression.

There are a large number of papers that describe the role of different signaling systems in pain models as described here. However, this report can be distinguished owing to the fact that a) the data set is very extensive, b) the effects observed are extremely robust and c) most importantly, the authors do not just present data on the role of FL/FLT3 but also go through the entire exercise of developing a novel "drug" that inhibits the FLT3 receptor and demonstrate its potential utility in the treatment of chronic pain.

There are some points that the authors should address.

- 1) In Fig 1a, b can the authors please describe the identity of these cells in more detail?
 - 2) DRG neurons are a heterogeneous group of cells. The authors say that they detected FLT3 in DRG neurons (supplementary material) but isn't clear which neurons express the receptor. It is important to establish this point. Moreover, the authors say that FL enhances the Ca response to different TRP receptors (TRPV1, TRPA1 and TRPM8). Again were all of these experiments carried out on the same neurons? Did the same neurons respond to all three TRP agonists or are these different groups of cells? Overall, it is important to clarify what types of DRG neurons are being identified or are responding to FL.
 - 3) Does the drug BDT001 only have effects in neuropathic pain? The pain models used are not really selective for neuropathic pain. Can the authors also present the effects of the drug on other pain models, particularly a model of inflammatory pain?
 - 4) In addition, the effects of the drug seem robust but there isn't really anything to compare it with in the context of this set of studies. How does the drug compare with something like morphine and/or a gabapentinoid run under the same conditions?
 - 5) Can the authors describe where else in the body the FL/FLT3 signaling system is expressed? Moreover, can they describe (eg from the literature) where in humans it is expressed?
- Minor point-
What was the source of the recombinant FL used and was it checked for endotoxin contamination.

Reviewer #2 (Remarks to the Author):

The manuscript of Rivat et. al. (NCMMS-17-11578) identifies the role of the cytokine FL and its corresponding receptor FLT3 in the development of neuropathic pain in mice. The authors show that the inhibition or absence of FLT3 using different approaches circumvent and/or reverse neuropathic pain (NP). To prove and verify the involvement of FL and FLT3 in the development of neuropathic pain they use numerous tools. They show expression of FL-positive cells in injured nerves, the ability of FL to induce thermal and mechanical hypersensitivity, lack of FL-induced hyperalgesia in Flt3 knock out mice and increased expression of NP-related biomarkers due to FL-administration. Furthermore, they present microfluometrical, electrophysiological and behavioral data showing increased TRPV1, TRPA1 and TRPM8 activity after FL-treatment which was absent in neurons of knock out mice. Behavioral investigation of Flt3 KO mice showed lack of NP development in CCI, SNI and SNL models. Using the CCI model they show that the increased mRNA expression and immunoreactivity of NP biomarkers like e.g. ATF or CSF1 was reversed in Flt3 KO mice. Furthermore, they verify in vivo and in vitro data using siRNA and shRNA. Finally, they identify a new molecule (BDT001) inhibiting FLT3 via structure-based in silico modeling and

tested its in vitro binding and ability to block FL-induced TRPV1 potentiation. CCI- or FL-induced NP symptoms were inhibited by BDT001 application. The study provides very detailed and convincing evidence for the involvement of FL and FLT3 in the development of NP and identifies a promising tool for the treatment of NP. However, there are some concerns which should be addressed prior to publication:

- The introduction states that "NP arises from aberrant functioning of somatosensory neurons present in DRG". This is incorrect because there are other forms of NP that do not involve DRG neurons. In addition, the animal models used here (CCI, SNI, SNL) solely represent NP associated with mechanical injury to peripheral nerves. This limits the conclusions because other (more clinically relevant) types of nerve injury (e.g. chemotherapy, infection, diabetes) are not represented by these models.
- The introduction also states that "Among [cytokines], with the notable exception of the cytokine FL and its cognate receptor FLT3, all the members of the class III receptor tyrosine kinase (RTK) family which comprises stem cell factor (SCF) receptor(c-Kit), colony-stimulating factor type-I (CSF1) receptor (CSF1R) and platelet-derived growth factor (PDGF) receptors (PDGFR) have been shown to be involved in normal nociception and/or pain". In view of the numerous non-immune factors involved in NP, this immediately raises the question whether the selective ablation of a single cytokine (FL) will ever be able to produce clinically relevant analgesia. This concern is further supported by their findings that "the capsaicin response was normal in cultures established from Flt3KO mice" (line 138), "BDT001 did not affect capsaicin-induced TRPV1 activation" (line 240), and "the compound at 10 μ M had no effect when TRPV1 potentiation was induced by NGF" (line 248). Again, this will limit the conclusions regarding clinical efficacy and indicates that a journal addressing a specialized audience may be more suitable for the rather narrow scope of this manuscript.
- Numerous antibodies are used but control experiments to show their specificity are missing (with the exception of FLT3 in suppl. Fig. 2). This is a growing problem in the scientific literature and must be addressed (see e.g. Nature 2015;521:274).
- It is not clear what the authors mean by "gene" (e.g. lines 119, 121, 123, 285; Fig. 1). In the methods they describe RNA extraction for RT-PCR. Also, they state "gene knock down by siRNA" (line 602). How can siRNA influence a gene?
- In Fig. 1c the authors showed an increased percentage of withdrawal 5 h after FL-injection. Did the authors check time points earlier than 5 h or is there any knowledge about the onset of FL-induced hypersensitivity and increased gene expression?
- Calcium imaging and patch clamp experiments in Fig. 2 show a rather rapid and direct way of action of FL, since the tachyphylaxis of TRP channels is reversed immediately after FL application. This cannot be explained by changes of expression levels, but possibly by phosphorylation events. Please comment on this.
- On page 4 line 135 the authors state that "FL alone had no effect on basal $[Ca^{2+}]_i$ ", however in Fig. 2b the second column (only FL application (0.56 nM)) shows almost the same ΔF peak as capsaicin alone which was shown to induce a calcium influx. This is contradictory, please comment on this. Additionally, it is not clear to which capsaicin application the response amplitudes were normalized (Peak 1 could probably be excluded).
- The finding that "the capsaicin response was normal in cultures established from Flt3KO mice" (line 138) is in contrast to "The mRNA expression of ... TrpV1 ... in DRG ... were reduced after Flt3 deletion" (line 174). Please comment.
- Please describe the stimulation protocol for TRPA1 and TRPM8. Was it the same as for TRPV1?
- Why did the authors use different concentrations of capsaicin and FL in their patch clamp experiments?
- Please also state the number of animals from which the cells were taken for measurements in Fig. 2b-f.
- Suppl. Fig. 2b: Did the authors also check other FL concentrations? Possibly the missing effect was due to a wrong dose? It is not clear for the reviewer why systemic FL should not work like near nerve injection (unless FL is not able to permeate the blood vessel wall), especially since it was shown in Fig. 6d that systemic intrathecal FL injection induced mechanical pain hypersensitivity.

- Suppl. Fig. 2c: Please provide also the appropriate controls for this experiment.
- For consistency of data presentation in Fig. 3 it would be good to show the time course for Fig. 3b too, as it was done for the other behavioral tests in this Fig. However this is a minor point.
- Why did the authors choose the two different time points (3 days post CCI and 14 days post CCI) to investigate mRNA expression and immunoreactivity for neuropathic pain markers?
- Page 5 lines 181 and 182: The authors state that NP-induced upregulation of mRNA of Csf1 ligand and CsfR is diminished in Flt3 KO mice. However, injection of CSF1 restored mechanical hyperalgesia in KO mice. How does this work if Csf1 receptor is downregulated? Please comment on this.
- Page 6 lines 202 and 203: The authors state that "GFP expression was restricted to DRG neurons." But in Suppl Fig. 4a and b the expression of anti-Flt3 shRNA and GFP is shown in DSC. Please explain the discrepancy.
- Please state the number experiments for Suppl. Fig. 5.
- Page 6 lines 234-236: It is stated that "BDT001 inhibited FL-induced FLT3 phosphorylation in leukemia-derived RS4-11 cells, also measured by TR-FRET (Supplementary Fig. 5)...". However in Suppl. Fig. 5 BDT001 was not investigated but another inhibitor AC220. Please comment on that.
- Please discuss long-term effects of FLT3 inhibition as it was shown to be involved in proliferation and differentiation of hematopoietic cells. Is there any influence on lymphocyte development?
- The reference list is divided into 2 parts by the method section (Ref. 1-43 on pages 10-13; Refs. 44-52 on pages 22-23).
- In the method section describing calcium imaging experiments it is stated that DRG neurons were loaded with 5 mM Fura-2 AM. This Fura concentration seems to be very high. Usually concentrations between 3 and 5 μ M are used in the literature.
- Page 22, line 834/835: It is stated that 2.9 commercially available drug-like compounds were filtered. The authors probably meant 2.9 million compounds, since this is stated in Fig. 5c.
- Suppl. Fig. 6, page 7 lines 182-184: "The ortho-phenol moiety seems to be mandatory for FLT3 binding since its deletion (compounds 68, 74, 77), displacement to the meta position (compound(s) 62), or methylation (compound 69) increases IC50 values. However, compound 69 displays the 3rd smallest IC50 value of all investigated compounds, right after compounds 66 and 75 (mentioned on page 6 being highly effective in inhibiting FL binding). Why do the authors conclude an increase of IC50 value for compound 69?"
- The grammar and style of the supplementary methods is poor. It is hard to follow and there is an inconsistency in spelling (μ L vs μ l, room temperature vs Room temperature, simple present passive vs simple past passive...), spelling errors and misleading verbs. Please revise this section.

Reviewer #3 (Remarks to the Author):

In this study, the authors define the role of Flt3 in the development of maintenance of nerve injury-induced mechanical hypersensitivity. They propose that during nerve injury, hematopoietic cells migrate to the site of injury, where they release the cytokine FL. This acts on the Flt3 receptor to produce mechanical and thermal pain hypersensitivity. Downregulating the expression of Flt3 alleviates the hypersensitization, as is the complete deletion of the Flt3 gene. Finally, they validate a negative allosteric modulator of Flt3 and demonstrate that it can alleviate the symptoms of nerve injury-induced mechanical and heat hyperalgesia.

They do this through an elegant combination of approaches including behavioral testing, Ca imaging, electrophysiology, the use of viruses delivered in vivo, and in silico screening of small molecules. The study is complete, going from the identification of a pathological mechanism, to the development of a novel drug with the potential to block this pathological step and alleviate pain after nerve injury.

Overall I believe this study has its place in the prestigious journal Nature Communications. The study is novel and its impact could be important.

My critique is divided in minor and major concern. I believe there is major work needed for the

resubmission, but believe the authors can provide that.

More important concerns:

1. Flt3 is also expressed in non neuronal cells that could be around the site of FL release. How do the authors determine whether the sensitization effects of FL injection are due activation of the neuronal Flt3 receptor and not the result of mast cell degranulation which would then activate nociceptors? The Flt3 ko experiments are performed in global KO mice. If the authors know which subset of sensory neurons normally express the Flt3 receptor, they can use an AAV injection of the wildtype FLT3 receptor in the KO mouse (either intra-sciatic injection) or intraperitoneal injection at P1-P3. The AAV can express Flt3 under a neuronal promoter (synapsin). Alternatively, the authors can cross the global KO mice to a nociceptor cre driver line (Trpv1-cre or Nav1.8-cre), then inject a virus carrying the flexed Flt3 receptor.
2. Why are the effects of FL injection measured only 5 hours post injection? Why not test the mice earlier?
3. For figure 2b, the authors describe that the application of FL had no effect on basal $[Ca^{2+}]_i$. but the graph shows the rise in $[Ca]_i$ induced by FL is comparable to that produced by application of capsaicin. I wouldn't say that FL has no effect.
4. Supplementary figure 2a. What is the sample number? It is surprising that an intra-sciatic nerve injection does not produce any pain. Can the authors comment on that?
5. Why are the time courses different in a and b? this should be made more consistent.
6. The i.v. dose is more dilute than the intra-sciatic injection. There is no comparison. The same dose should be tested.
7. The quantification of the validation experiments in figure suppl 3a and b is difficult to understand. Why didn't the authors simply use the Western blot approach like they did for figure S4c?
8. An MAJOR concern of the methodology that is of concern is the test used to assess mechanical nociception. (Fig. 4 a and Fig S3c). Simply recording withdrawal responses is not an indication of nociception. These are spinal reflexes and may not even be related to pain. The effective way to record this is to examine the filaments that cause a nociceptive-like response, during which the mouse will not only withdraw its paw from the filament, but also shake it, or lick it, or try to bite the filament. A proper approach for this type of pain vs reflex assessment is described by Ducourneau and Dallel. Without this type of assessment, it is NOT POSSIBLE to claim a withdrawal is a nociceptive response. Even if withdrawals occur with small filaments, or if the percent of withdrawals with the same filament increase, it is still not an indication of more pain. The only indication of more pain is if the mouse starts displaying signs of nociception (flicking, licking,...) occur with smaller filaments.
9. The data with the BDT001 compound is almost premature in this paper. There is a lot of characterization of this compound which could have been done in a separate manuscript. It is not clear to me what is the consequence of BDT001 inhibiting EphB4 receptor binding by 23%. This percent inhibition is not insignificant. How do the authors make sure the drug is selective to their target? Other Flt3 blockers certainly do have off-target effects. All this could be in a separate paper.
10. It is not clear to me what is the effect of BDT001 alone on the basal Ca level in DRGs.
11. Putting in a mixture of techniques, including virtual screening, essentially makes reviewers unable to judge the whole manuscript. I am unable to tell whether the virtual screening was done using thorough methods and if proper controls were added or not. It is my opinion, nonetheless, that this compound should be more thoroughly validated elsewhere, in its own paper, before being incorporated here.

Overall, the effect on mechanical pain sensing cannot really be interpreted because the appropriate test was not run. Even the effect on the Hargreaves' apparatus should be quantified in terms of the minimum latency to not only evoke a paw withdrawal, but a paw withdrawal followed by shaking or licking of the paw. Otherwise these behaviors remain spinal reflexes.

The role of neuronal Flt3 should be demonstrated. The expression of Flt3 should be confirmed not only by Western blot of DRGs, but also by showing that it is the neuronal Flt3 that is responsible for the effect.

Minor comments:

Can the authors define what FL stands for?

What could control FLT3 expression level? One would think that nerve injury would upregulate FLT3, which would later be activated by FL. Why would neurons express FLT3 under normal conditions?

Discussion page 8 line 289: FLT3 modulates the cytokine CSF1, ...

What does that sentence mean? Modulates what aspect of CSF1? The expression level? The localization? Please be more precise.

Introduction page 2 line 43: when describing the high social costs, it would be useful to give the readers an idea of what these numbers are. Readership is from mixed backgrounds and these are not obvious facts to everyone.

Page 2 line 55-57: this is a broad statement. Some references are necessary to back this claim. I have not heard the term chronicization. Most people in the field use chronification. For the sake of consistency, could you please use chronification?

Line 71L "secrete sensitizers". This is another vague term being used. Give examples (with references) of molecules being released please, and avoid using vague terms.

Page 3 line 83: the reference you want to cite is #16, not 10,11.

Results:

Based on the single cell PCR experiments of Usoskin et al., Nature Neuroscience (2015), Flt3 expression appears to be the highest in proprioceptors. Can the authors comment on this?

Page 3 line 107: von Frey is with a small v

Page 4 line 147: "...recapitulates some of the molecular.."

Page 6 line 206: it is not an "irrelevant shRNA", but rather a non-targeting shRNA. Not shRNA is irrelevant.

Methods: define the anesthetic used (line 518

Line 520: please edit the references, there is no reference #45

Line 544: the make mice argument should be in the first paragraph of methods.

Line 556: the Dixon method: please provide the proper reference for this paper.

Line 561: these tests are not measures of nociception.

Line 576: the cumulative reaction time: should be labeled simply as Duration of responses.

Line 578: it is not clear to me why anyone would inject 4 times capsaicin in the mouse paw. I can understand it is to mirror the in vitro data, but capsaicin produces quite a painful reaction in the mouse that can trigger central sensitization. Thus recruiting additional variables that have nothing to do with the processes going on in the sensory neuron.

Line 584: drug delivery. Please provide more information on catheterization: size, what is the catheter made of? Polyethylene? What is the internal diameter, outer diameter?; where was it anchored? Where was it inserted from? Where was it implanted? Text only says dorsal subcutaneous space.

Line 592: a total of 10 ul of drug was injected. This seems quite high. If one was to inject 10 ul of lidocaine intrathecally, the animal would stop breathing because 10 ul can reach the brainstem. Doses used in mice are typically no more than 5 ul.

Line 597: label as intra-sciatic injection. Was it the left or the right sciatic nerve bundle?

Gene knockdown experiments:

siRNA approach: cells are overexpressing Flt3, thus at various levels. Using a knockdown approach like this makes it difficult to properly quantify how much knockdown you get. If the experiment cannot be done on the native cells (DRGs), then the authors should use a cell line endogenously expressing Flt3 and validate the siRNA knock down.

Line 653: the cell dissociation method should be described before the electrophysiology or the Ca imaging.

Line 685: what is Locks? Please define.

Line 723: virus transfection: please organize better the justification of lines 724-731.

Line 744: AAV not AVV

Line 756: were41?

Line 759: what is mowiol?

Virtual screening: seems premature to incorporate in this paper. The validation should be done elsewhere.

Figures:

Figure 1c: you cannot use in the legend the term Mechanical Pain Sensitivity. Your test does not allow you to distinguish if the withdrawal was a reflex or if it was a nociceptive (not pain) response.

Figure 1d. label should be duration of response, not reaction time. Reaction time may indicate the time it took for the mouse to respond. Similarly, "paw withdrawal threshold" is typically used for von Frey tests. "latency (sec)" is more appropriate here.

Furthermore: the cold effect 1.5 sec to 2.5 sec is not a dramatic increase, despite being significant.

The heat latencies are quite low. The bulb intensity should be adjusted such that baseline values of ~10 seconds are needed. Then you would test the FL effect. You are near saturation latencies.

Going from 5.5 sec to 4 sec is not dramatic, despite being significant.

Figure 2c: there is some additional information missing here. What happens to the Flt3 KO in response to capsaicin alone?

It seems the authors already have a patent on Flt3 receptor blockers in the treatment of pain. This makes me wonder if the data has been published elsewhere (patent).

Review Nature communications NCOMMS-17-11578-T

Reviewer #1: (Remarks to the Author):

This report describes a very extensive and impressive series of experiments designed to highlight the central role of the FL/FLT3 cytokine signaling system in the establishment and maintenance of neuropathic pain as assessed using rodent (mice) pain models. FLT3 activation in DRG neurons resulted in rapid pain due to transactivation of TRP channels as well as longer lasting effects due to regulation of gene expression. There are a large number of papers that describe the role of different signaling systems in pain models as described here. However, this report can be distinguished owing to the fact that a) the data set is very extensive, b) the effects observed are extremely robust and c) most importantly, the authors do not just present data on the role of FL/FLT3 but also go through the entire exercise of developing a novel “drug” that inhibits the FLT3 receptor and demonstrate its potential utility in the treatment of chronic pain.

We thank the reviewer for its positive appreciation of our manuscript. We provide point-by-point answers to all the comments and indicate the additional experiments we performed to clarify all the points raised.

There are some points that the authors should address.

1) In Fig 1a, b can the authors please describe the identity of these cells in more detail?

We have added a Supplementary Figure 1 to characterize in more details the identity of the FL-positive/CD45+ cells by using specific markers of immune cells populations. The manuscript was modified accordingly. Further analysis showed that 60% of the FL+ population were CD11b+ (a marker of myeloid cells), but none was CD68+ (a marker of macrophage). No co-localization was found with CD3 (a marker of T lymphocytes). Thus, these added results suggest that the FL+ population present in the CCI-injured nerve might belongs to the monocytes, neutrophils and/or natural killer immune subsets (page 3, in Result paragraph 1).

2) DRG neurons are a heterogeneous group of cells. The authors say that they detected FLT3 in DRG neurons (supplementary material) but it isn't clear which neurons express the receptor. It is important to establish this point. Moreover, the authors say that FL enhances the Ca response to different TRP receptors (TRPV1, TRPA1 and TRPM8). Again were all of these experiments carried out on the same neurons? Did the same neurons respond to all

three TRP agonists or are these different groups of cells? Overall, it is important to clarify what types of DRG neurons are being identified or are responding to FL.

We acknowledge that we didn't describe very thoroughly the expression pattern of FLT3 receptor expression in the DRG. In fact, we tested all of the FLT3 antibodies commercially available (listed below). However, in our hands they exhibited inconsistent results regarding one of the known positive controls i.e cerebellum (positive control) or in the *Flt3*^{KO} mice (negative control) and thus were not helpful for resolving this question.

Nevertheless, our *in situ* hybridization results (Supplementary Fig. 3) suggest a neuron-specific expression over a broad range of neuronal cell diameters in the DRG, that was absent in *Flt3*^{KO} mice. These *in situ* results are in broad agreement with a single cell RNAseq analysis carried out on mouse DRG neurons by Usoskin et al., 2015 (see reference #26 in the revised version of manuscript), that showed expression of *Flt3* transcripts in different DRG subpopulations, such as myelinated sensory neurons (proprioceptors, low threshold mechanoreceptors, A δ -fibers nociceptors) and at a lower level a population of non-peptidergic C-fiber nociceptors.

Regarding TRP channels, in physiological conditions, TRP channels are expressed in nociceptors, mainly in peptidergic neurons for TRPV1 and TRPM8 and in IB4 positive non-peptidergic neurons for TRPA1^{1,2}.

Our functional studies clearly established that direct applications of FL potentiates TRP function *in vitro* and *in vivo*, and this effect is abolished in *Flt3*^{KO} mice. Our data (Fig. 2) showing a rapid action (seconds) of FL on TRP channels *in vitro* strongly suggest that at least a part of FLT3 expressing neurons express one of the TRP channel subtypes investigated i.e TRPV1, TRPA1 or TRPM8. Concerning calcium imaging studies, the different TRP agonists have been tested on individual neurons over different experiments. However, a substantial overlap between TRPV1 and TRPA1 as well as between TRPV1 and TRPM8 expressing sensory neurons in adult mouse DRG has been well established¹.

N.B.: The antibodies tested are : ***Anti-FLT3 antibodies tested in immunohistochemistry: Anti-Flt3 Abcam ab73019 (rat, clone 12B6), anti-Flt3 Cell Signaling 3462 (rabbit, clone 8F2), anti-Flt3 Santa Cruz sc-20733 (rabbit, clone H300), anti-Flt3 Santa Cruz sc-480 (rabbit, clone S18), anti-Flt3 Santa Cruz sc-101343 (mouse, clone 8H5), anti-Flt3 R&D System MAB7681 (rat, clone 113308), anti-Flt3 Millipore 06-647 (rabbit), anti-Flt3 Millipore 06-646 (rabbit), anti-Flt3 GeneTex GTX53112 (rat, clone 12B6), anti-Flt3 Proteintech 21049-AP (rabbit) and anti-Flt3 PE BD Pharmagen 553842 (rat). None of these antibodies gave a reliable immunostaining signal, which was present on sections prepared from both wild-type and *Flt3*^{KO} mice.***

3) Does the drug BDT001 only have effects in neuropathic pain? The pain models used are not really selective for neuropathic pain. Can the authors also present the effects of the drug on other pain models, particularly a model of inflammatory pain?

We have added new experiments to test BDT001 in inflammatory pain models. We used the chronic inflammatory pain induced by CFA injection. BDT001 does not alter responses to chronic inflammatory pain in the CFA model (Fig. 6i and page 8, paragraph 2).

4) In addition, the effects of the drug seem robust but there isn't really anything to compare it with in the context of this set of studies. How does the drug compare with something like morphine and/or a gabapentinoid run under the same conditions?

As suggested by the referee, these experiments have now been done by comparing effects of BDT001 and a standard of care, the gabapentinoid pregabalin (Fig. 6 f, g and page 8, paragraph 2). The results show a longer duration of action for BDT001, compared to pregabalin.

5) Can the authors describe where else in the body the FL/FLT3 signaling system is expressed? Moreover, can they describe (eg from the literature) where in humans it is expressed?

The literature data indicate that FLT3 is highly expressed in hematopoietic organs, such as spleen, thymus, peripheral blood, bone marrow. *Flt3* transcripts are also expressed in various regions of the adult brain. This is now specified in the Introduction (page 3, paragraph 2). We have added a panel in Supplementary Fig. 3C to indicate that the FLT3 expression level is much higher in a typical brain region, the cerebellum, than in peripheral sensory system.

6) Minor point: What was the source of the recombinant FL used and was it checked for endotoxin contamination.

Recombinant rh-FLT3-L was produced in E. Coli Rosetta (DE3) strain (Novagen) in our laboratory using the pET15b-rhFL plasmid according to the protocol described by Verstraete et al in Protein J (2009) with some minor modifications (see Supp. Methods, Chemicals and Reagents Materials and Methods, page 21, paragraph 4). The rh-FL was checked for endotoxin content using the Pyrogen Recombinant Factor C endotoxin detection assay from LONZA (Walkersville MD, USA) and was found free of endotoxins. This is now specified page 21, paragraph 4.

Reviewer #2 (Remarks to the Author):

The manuscript of Rivat et. al. (NCMMS-17-11578) identifies the role of the cytokine FL and its corresponding receptor FLT3 in the development of neuropathic pain in mice. The authors show that the inhibition or absence of FLT3 using different approaches circumvent and/or reverse neuropathic pain (NP). To prove and verify the involvement of FL and FLT3 in the development of neuropathic pain they use numerous tools. They show expression of FL-positive cells in injured nerves, the ability of FL to induce thermal and mechanical

hypersensitivity, lack of FL-induced hyperalgesia in *Flt3* knock out mice and increased expression of NP-related biomarkers due to FL-administration. Furthermore, they present microfluometrical, electrophysiological and behavioral data showing increased TRPV1, TRPA1 and TRPM8 activity after FL-treatment which was absent in neurons of knock out mice. Behavioral investigation of *Flt3* KO mice showed lack of NP development in CCI, SNI and SNL models. Using the CCI model they show that the increased mRNA expression and immunoreactivity of NP biomarkers like e.g. ATF or CSF1 was reversed in *Flt3* KO mice. Furthermore, they verify in vivo and in vitro data using siRNA and shRNA. Finally, they identify a new molecule (BDT001) inhibiting FLT3 via structure-based in silico modeling and tested its in vitro binding and ability to block FL-induced TRPV1 potentiation. CCI- or FL-induced NP symptoms were inhibited by BDT001 application. The study provides very detailed and convincing evidence for the involvement of FL and FLT3 in the development of NP and identifies a promising tool for the treatment of NP.

We thank the reviewer for its valuable comments and constructive analysis of our manuscript. We provide point-by-point answers to all the comments and indicate the additional experiments we performed to clarify all the points raised.

However, there are some concerns which should be addressed prior to publication:

1) The introduction states that “NP arises from aberrant functioning of somatosensory neurons present in DRG“. This is incorrect because there are other forms of NP that do not involve DRG neurons. In addition, the animal models used here (CCI, SNI, SNL) solely represent NP associated with mechanical injury to peripheral nerves. This limits the conclusions because other (more clinically relevant) types of nerve injury (e.g. chemotherapy, infection, diabetes) are not represented by these models.

We have taken into consideration the reviewer’s valuable criticism that there are other neuropathic pain conditions not caused by mechanical injury to peripheral neurons. Firstly, we now limit our conclusions to peripheral neuropathic pain (PNP instead of NP). Secondly, we now provide results obtained in a model of generalized PNP, induced by chemotherapy, showing that *Flt3*-deficient mice do not develop mechanical hypersensitivity after treatment with oxaliplatin (Fig 3g) and in Materials and Methods “Chronic pain models” section.

2) The introduction also states that “Among [cytokines], with the notable exception of the cytokine FL and its cognate receptor FLT3, all the members of the class III receptor tyrosine kinase (RTK) family which comprises stem cell factor (SCF) receptor(c-Kit), colony-stimulating factor type-1 (CSF1) receptor (CSF1R) and platelet-derived growth factor (PDGF) receptors (PDGFR) have been shown to be involved in normal nociception and/or pain“. In view of the numerous non-immune factors involved in NP, this immediately raises the question whether the selective ablation of a single cytokine (FL) will ever be able to produce clinically relevant analgesia. This concern is further supported by their findings that “the capsaicin response was normal in cultures established from *Flt3*KO mice” (line 138), “BDT001 did not affect capsaicin-induced TRPV1 activation” (line 240), and “the compound at 10 μ M had no effect

when TRPV1 potentiation was induced by NGF” (line 248). Again, this will limit the conclusions regarding clinical efficacy and indicates that a journal addressing a specialized audience may be more suitable for the rather narrow scope of this manuscript.

We would like to clarify these points. Our results do not concern “the selective ablation of a single cytokine (FL)” but the modulation of FLT3, its receptor. In addition, inhibiting FLT3 or ablating FLT3 expression does not, in any case, produce analgesia, but prevents or reverses hyperalgesia and pain sensitization after nerve lesion. Indeed, our results show that FLT3 has no role in normal nociception because:

- FLT3 stimulation does not activate nociceptors in physiological conditions (this is now specified in Fig. 2b, and text page 4, paragraph 3),
- Flt3-deficient mice have normal responses to capsaicin (now specified in Fig. 2c and text page 4, paragraph 3),
- Flt3-deficient mice have normal responses to mechanical, thermal (Supplementary Fig. 2c-f) or chemical nociception (new Supplementary Fig. 4a) and normal response to inflammatory pain (new Supplementary Fig. 4a, b),
- BDT001 does not affect normal nociception (Fig. 6j) nor inflammatory pain (Fig. 6i; new data).

However, peripheral neuronal FLT3 activation is *necessary* to induce PNP and its inhibition sufficient to reverse neuropathic hypersensitivity and many of the associated molecular changes. So FLT3 appears to be a key upstream modulator (trigger) of PNP physiopathology.

Thus, we respectfully disagree with this Reviewer’s skepticism that inhibition of FLT3, which is able to treat neuropathic pain, will have clinical relevance. For these reasons we think that our findings are of general interest.

3) Numerous antibodies are used but control experiments to show their specificity are missing (with the exception of FLT3 in suppl. Fig. 2). This is a growing problem in the scientific literature and must be addressed (see e.g. Nature 2015;521:274).

The antibodies that were used in the present experiments are commercially available and have been widely used in similar experiments, for which there is an abundant literature. There are no discrepancies between the present data and those published regarding the proportions of identified neurons in WT animals (Supplementary Fig. 2a, b) or the distribution of markers in the DRG or DSC of CCI-WT animals (Fig 3i-l). The only antibody that had not been previously used was the anti-FLT3 antibody, and this antibody was validated for Western blotting by using *Flt3*-deficient mice (Supplementary Fig. 3c).

- It is not clear what the authors mean by “gene” (e.g. lines 119, 121, 123, 285; Fig. 1). In the methods they describe RNA extraction for RT-PCR. Also, they state “gene knock down by siRNA” (line 602). How can siRNA influence a gene?

We have clarified these points in the text using the expressions “gene expression levels” or “gene transcripts” at many places in the manuscript.

- In Fig. 1c the authors showed an increased percentage of withdrawal 5 h after FL-injection. Did the authors check time points earlier than 5 h or is there any knowledge about the onset of FL-induced hypersensitivity and increased gene expression?

Fig. 1c shows clearly that FL-induced hyperalgesia is dose and time-dependent. Since at the highest dose tested (500 ng), the maximal effect was obtained 5 h after FL-injection (personal data), this time point was retained for subsequent experiments at the same dose.

- Calcium imaging and patch clamp experiments in Fig. 2 show a rather rapid and direct way of action of FL, since the tachyphylaxis of TRP channels is reversed immediately after FL application. This cannot be explained by changes of expression levels, but possibly by phosphorylation events. Please comment on this.

We agree that such a short TRP activation by FL in cultured DRG neurons is likely to involve phosphorylation, as demonstrated for other TRP activators (see for example³). This is now specified in the text. Nevertheless, as we show in Fig. 1g, FLT3 activation by FL is also likely to induce long-term effects through TRP gene upregulation.

- On page 4 line 135 the authors state that “FL alone had no effect on basal $[Ca^{2+}]_i$ ”, however in Fig. 2b the second column (only FL application (0.56 nM) shows almost the same ΔF peak as capsaicin alone which was shown to induce a calcium influx. This is contradictory, please comment on this. Additionally, it is not clear to which capsaicin application the response amplitudes were normalized (Peak 1 could probably be excluded).

The Reviewer is correct; this was an omission. Figure 2b has now been corrected and shows that FL alone has no effect on basal $[Ca^{2+}]_i$.

Pulses of capsaicin were applied 4 times at 2 min-intervals and capsaicin with drugs was added after the third pulse. The normalization was done by calculating the ratio of Peak4/Peak3. This now added in the Material and Methods “Calcium imaging” section.

- The finding that “the capsaicin response was normal in cultures established from Flt3KO mice” (line 138) is in contrast to “The mRNA expression of ... TrpV1 ... in DRG ... were reduced after Flt3 deletion” (line 174). Please comment.

This was poorly written. In fact, Fig. 3h shows that there was no difference in TrpV1 gene expression between WT and KO mice in non-injured animals, from which DRG cultures were established; it is therefore not surprising that the capsaicin response was not altered. However, increases in mRNA expression of Atf3, NpY, TrpV1 and Gfap in DRG and Iba1,

Cd11b and Gfap in DSC were abrogated or diminished after Flt3 deletion in nerve injured animals. The text has been corrected accordingly (page 6, paragraph 1).

-Please describe the stimulation protocol for TRPA1 and TRPM8. Was it the same as for TRPV1?

The same temporal stimulation protocol was used for the three TRPs. This is now added to the Materials & Methods “Calcium imaging” section.

- Why did the authors use different concentrations of capsaicin and FL in their patch clamp experiments?

First, many previously published patch-clamp experiments have used a range of concentrations between 0.5-1 μ M to induce capsaicin currents in current clamp experiments, with better kinetics at 1 μ M. Second, we wanted to have saturating concentrations of FL (between 5.6 and 10 nM) to analyze this effect with dynamic kinetics that is more rapid with patch clamp than with calcium imaging.

- Please also state the number of animals from which the cells were taken for measurements in Fig. 2b-f.

Calcium imaging have been done for at least 3 different animals for each experimental conditions. This is now specified in Materials and Methods, “Calcium imaging” section, page 18.

- Suppl. Fig. 2b: Did the authors also check other FL concentrations? Possibly the missing effect was due to a wrong dose? It is not clear for the reviewer why systemic FL should not work like near nerve injection (unless FL is not able to permeate the blood vessel wall), especially since it was shown in Fig. 6d that systemic intrathecal FL injection induced mechanical pain hypersensitivity.

We have repeated this experiment using a dose of 5 mg/100 μ l for the intravenous injection. The result was again negative with this high dose. This is now revised in the new Suppl. Fig. 3b.

-why systemic FL should not work like near-nerve injection?

FL nerve injection is done to bypass the peripheral nerve-blood barrier. It is not a “near nerve injection”, but an “intra-nerve injection”.

In Fig. 6d it was not “systemic” intrathecal injection, but localized intrathecal injection. This intrathecal injection applies the solution into subarachnoid space which is contiguous with the dorsal roots.

- Suppl. Fig. 2c: Please provide also the appropriate controls for this experiment.

We have added a positive control to the Western blot, showing the detection of FLT3 protein in one brain tissue i.e. cerebellum and the absence of the FLT3 band in *Flt3*-null mutant tissue (Supplementary Fig. 3c).

- For consistency of data presentation in Fig. 3 it would be good to show the time course for Fig. 3b too, as it was done for the other behavioral tests in this Fig. However this is a minor point.

We have not performed a complete time-course. To limit the number of animals used in our investigation, we performed the Hargreaves test and the von Frey test in the same groups of animals. Hence, the repetition of nociceptive tests in nerve injured-animals was limited here to avoid a potential pain sensitization produced by the repeated application of the nociceptive stimulus itself.

- Why did the authors choose the two different time points (3 days post CCI and 14 days post CCI) to investigate mRNA expression and immunoreactivity for neuropathic pain markers?

In mice, the first week corresponds to the period of development of the peripheral and central hypersensitivity and the installation of the PNP pathology. Thereafter, the pathological state is stable and maintained (see Fig. 3a; Fig. 4a,g). By choosing these two time points, we can therefore assess both the changes during the dynamic process and the stable state.

- Page 5 lines 181 and 182: The authors state that NP-induced upregulation of mRNA of *Csf1* ligand and *CsfR* is diminished in *Flt3* KO mice. However, injection of CSF1 restored mechanical hyperalgesia in KO mice. How does this work if *Csf1* receptor is downregulated? Please comment on this.

Our description was not clear. Fig. 3m shows that the injury-induced increases of CSF1 and CSF1R expression were attenuated in *Flt3 null* mutant mice, but CSF1 and CSF1R were normal in sham-operated *Flt3* KO mice. So, this is not surprising that *Flt3* KO mice respond to CSF1 in Fig. 3p. The text has been modified accordingly (page 6, paragraph 1).

- Page 6 lines 202 and 203: The authors state that "GFP expression was restricted to DRG neurons." But in Suppl Fig. 4a and b the expression of anti-*Flt3* shRNA and GFP is shown in DSC. Please explain the discrepancy.

AAV9 cell transduction, as reported by GFP, is limited to neurons in the DRG (co-staining with Tuj1) and was excluded from other cellular populations of the DRG such as CD45+

resident macrophages and GFAP+ satellite glial cells (Fig. 4d). GFP staining in the dorsal spinal cord labeled fiber-like processes, which almost certainly originate from sensory afferent projections, and was absent from NeuN+ neuron cell bodies and CD45+ microglia (Supp. Fig. 6a, b).

- Please state the number experiments for Suppl. Fig. 5.

It is now in Suppl. Fig. 6. and the experiments were done in triplicate.

- Page 6 lines 234-236: It is stated that “BDT001 inhibited FL-induced FLT3 phosphorylation in leukemia-derived RS4-11 cells, also measured by TR-FRET (Supplementary Fig. 5)...”. However in Suppl. Fig. 5 BDT001 was not investigated but another inhibitor AC220. Please comment on that.

In this Figure, now Suppl Fig.7, we describe the two in vitro assays (panel a: binding of FL to FLT3 in HEK cells, panel c: FL-induced phosphorylation of FLT3 in RS4-11 cells) that we have set-up to evidence extracellular FLT3 inhibition by competitive binding (panel b) and blockade of FL-induced FLT3 phosphorylation (panel d). Since no extracellular FLT3 inhibitor was known when the assay was developed, validation of the phosphorylation assay was done using an intracellular non-selective FLT3 inhibitor (AC220). Once the assay validated (Suppl Fig. 7), it has been later used to screen potential extracellular FL3 inhibitors and identify BDT001 (Fig. 5d-g).

- Please discuss long-term effects of FLT3 inhibition as it was shown to be involved in proliferation and differentiation of hematopoietic cells. Is there any influence on lymphocyte development?

***Flt3*-deficient mice have no gross abnormalities in their hematopoietic system: they have normal absolute numbers of mature myeloid and lymphoid cells and thymocytes subpopulations in the spleen and of mature monocyte, granulocyte and erythrocyte numbers in the bone marrow; however, they show a 33% reduction of early B -cell progenitors, whereas the number of more mature B cells is normal⁴. We summarize these literature findings in the text by stating that *Flt3*-deficient mice have a normal mature hematopoietic system, even though some discrete populations of cell progenitors are reduced (page 3, paragraph 2).**

- The reference list is divided into 2 parts by the method section (Ref. 1-43 on pages 10-13; Refs. 44-52 on pages 22-23).

It is now corrected

- In the method section describing calcium imaging experiments it is stated that DRG neurons were loaded with 5 mM Fura-2 AM. This Fura concentration seems to be very high. Usually concentrations between 3 and 5 μ M are used in the literature.

This was a mistake, now corrected: the concentration was 2.5 μ M.

- Page 22, line 834/835: It is stated that 2.9 commercially available drug-like compounds were filtered. The authors probably meant 2.9 million compounds, since this is stated in Fig. 5c.

We acknowledge the Reviewer for pinpointing this discrepancy. Indeed, 2.9 million compounds have been screened in silico. The manuscript has been modified accordingly page 22 in the 'Virtual screening' section of the 'Online Methods' section.

- Suppl. Fig. 7, page 7 lines 182-184: "The ortho-phenol moiety seems to be mandatory for FLT3 binding since its deletion (compounds 68, 74, 77), displacement to the meta position (compound(s) 62), or methylation (compound 69) increases IC₅₀ values. However, compound 69 displays the 3rd smallest IC₅₀ value of all investigated compounds, right after compounds 66 and 75 (mentioned on page 6 being highly effective in inhibiting FL binding). Why do the authors conclude an increase of IC₅₀ value for compound 69?"

The reviewer is right in stating that the methoxy analog 69 is among the best binders. However, it only partially inhibits FLT3 (35% inhibition at 100 μ M) with respect to the phenol derivative 11 (66% inhibition 100 μ M). To ascertain the deleterious effect of methylating the phenol group, we have synthesized and tested the methoxy analog of BDT001. Its much weaker potency (IC₅₀ = 58 μ M vs. IC₅₀=17 μ M for BDT001) supports our original statement.

- The grammar and style of the supplementary methods is poor. It is hard to follow and there is an inconsistency in spelling (μ L vs μ l, room temperature vs Room temperature, simple present passive vs simple past passive...), spelling errors and misleading verbs. Please revise this section.

The manuscript has been fully edited by a native English speaker (Patrick Carroll). Modifications have been carried out throughout the text.

Reviewer #3 (Remarks to the Author):

In this study, the authors define the role of Flt3 in the development of maintenance of nerve injury-induced mechanical hypersensitivity. They propose that during nerve injury, hematopoietic cells migrate to the site of injury, where they release the cytokine FL. This acts on the Flt3 receptor to produce mechanical and thermal pain hypersensitivity.

Downregulating the expression of Flt3 alleviates the hypersensitization, as is the complete deletion of the Flt3 gene. Finally, they validate a negative allosteric modulator of Flt3 and demonstrate that it can alleviate the symptoms of nerve injury-induced mechanical and heat hyperalgesia.

They do this through an elegant combination of approaches including behavioral testing, Ca imaging, electrophysiology, the use of viruses delivered in vivo, and in silico screening of small molecules. The study is complete, going from the identification of a pathological mechanism, to the development of a novel drug with the potential to block this pathological step and alleviate pain after nerve injury. Overall I believe this study has its place in the prestigious journal Nature Communications. The study is novel and its impact could be important.

We thank the reviewer for its positive appreciation, valuable comments and constructive analysis of our manuscript. We provide point-by-point answers to all the comments and indicate the additional experiments we performed and the text alterations we made to clarify all the points raised.

My critique is divided in minor and major concern. I believe there is major work needed for the resubmission, but believe the authors can provide that.

More important concerns:

1. Flt3 is also expressed in non neuronal cells that could be around the site of FL release. How do the authors determine whether the sensitization effects of FL injection are due activation of the neuronal Flt3 receptor and not the result of mast cell degranulation which would then activate nociceptors? The Flt3 ko experiments are performed in global KO mice. If the authors know which subset of sensory neurons normally express the Flt3 receptor, they can use an AAV injection of the wildtype FLT3 receptor in the KO mouse (either intrasciatic injection) or intraperitoneal injection at P1-P3. The AAV can express Flt3 under a neuronal promoter (synapsin). Alternatively, the authors can cross the global KO mice to a nociceptor cre driver line (Trpv1-cre or Nav1.8-cre), then inject a virus carrying the flexed Flt3 receptor.

We agree with this Reviewer that it is essential to determine that the action of FL is exerted *via* neuronal FLT3 and not *via* mast cell degranulation and activation of nociceptors. The results already presented in the original Figure 4d strongly suggest that it is neuronal FLT3 that is responsible for the observed effects on the CCI model of PNP. Firstly, AAV9 cell transduction, as reported by GFP, is limited to neurons in the DRG (co-staining with Tuj1) and was excluded from other cellular populations of the DRG such as CD45+ resident macrophages and GFAP+ satellite glial cells (Fig. 4d). Secondly, GFP staining in the dorsal spinal cord labeled fiber-like processes, which most likely originate from sensory afferent projections, and was absent from NeuN+ neuron cell bodies and CD45+ microglia (Supp. Fig. 6a, b).

To make the demonstration even stronger, we performed new experiments, which allow us to definitively answer this question. Using the anti-Flt3-shRNA-AAV9 viral strategy, we performed an intrathecal FL injection in mice. FL-induced mechanical hyperalgesia was

strongly attenuated in mice that have previously received anti-Flt3 shRNA into DRG neurons, but not in mice receiving control shRNA (Fig. 4e, f). A description of these results is now added in the text (page 6, paragraph 2)

Thus, abrogating Flt3 expression uniquely in DRG neurons is sufficient to render FL ineffective.

2. Why are the effects of FL injection measured only 5 hours post injection? Why not test the mice earlier?

Fig. 1c shows clearly that FL-induced hyperalgesia is dose and time-dependent. Since at the highest dose tested (500 ng), the maximal effect was obtained 5h after FL-injection (personal data), this time point was retained for subsequent experiments at the same dose.

3. For figure 2b, the authors describe that the application of FL had no effect on basal $[Ca^{2+}]_i$ but the graph shows the rise in $[Ca^{++}]_i$ induced by FL is comparable to that produced by application of capsaicin. I wouldn't say that FL has no effect.

The reviewer is correct; this was an omission on our part. Figure 2b has been corrected and now shows clearly that FL alone has no effect on basal $[Ca^{++}]_i$

4. Supplementary figure 2a. What is the sample number? It is surprising that an intra-sciatic nerve injection does not produce any pain. Can the authors comment on that?

As requested by the authors, we now added the sample number for Supplementary Figure 3a (new Sup fig. 2a) (n=12). Regarding the effects of the intra-sciatic nerve injection on pain sensitivity, the saline injection did produce slight changes in the nociceptive threshold but they were not significant. We took numerous cautions to avoid damaging the nerve by using a 33G needle to disrupt the epineurium and then to insert a polyethylene tubing allowing to slowly inject the solution. The details of the method are now mentioned in the revised version of the manuscript. In addition, we evaluated the nociceptive threshold 5h after the injection. Testing immediately after the injection e.g. 30 min or less, would certainly result in significant changes of nociceptive threshold. Here we wanted to look at FL effects and avoid potential effects produced by the injection itself.

5. Why are the time courses different in a and b? This should be made more consistent.

We have changed Supplementary Fig. 3b to show the same time-course as in a.

6. The i.v. dose is more dilute than the intra-sciatic injection. There is no comparison. The same dose should be tested.

We have redone this experiment using a dose of 5 µg/100 µl for the intravenous injection. The result was again negative. This is now revised in the new Supp. Fig. 3b.

7. The quantification of the validation experiments in figure suppl 3a and b is difficult to understand. Why didn't the authors simply use the Western blot approach like they did for figure S4c?

We have performed new Western blot analysis in HEK293 cells overexpressing mouse FLT3 to assess the effect of anti-Flt3 siRNA on FLT3 expression. Results show a drastic down-regulation of FLT3 protein levels, as compared to a scrambled siRNA. This is now shown in Supplementary Fig. 5b, c.

8. An MAJOR concern of the methodology that is of concern is the test used to assess mechanical nociception. (Fig. 4 a and Fig S3c). Simply recording withdrawal responses is not an indication of nociception. These are spinal reflexes and may not even be related to pain. The effective way to record this is to examine the filaments that cause a nociceptive-like response, during which the mouse will not only withdraw its paw from the filament, but also shake it, or lick it, or try to bite the filament. A proper approach for this type of pain vs reflex assessment is described by Ducourneau and Dallel. Without this type of assessment, it is NOT POSSIBLE to claim a withdrawal is a nociceptive response. Even if withdrawals occur with small filaments, or if the percent of withdrawals with the same filament increase, it is still not an indication of more pain. The only indication of more pain is if the mouse starts displaying signs of nociception (flicking, licking,...) occur with smaller filaments.

We totally agree with this Reviewer and this point is critical. To address this issue, we took into consideration the recommendation made by the Reviewer. Hence, we performed additional experiments in which we measured paw withdrawal thresholds as in the initial manuscript, but also we determined a pain score. We did not use the method of assessment described in Ducourneau's study because this has been done in rats and the number of mechanical stimulations performed in this study was quite high and it can produce pain sensitization as reported in the publication by Ma and Woolf⁵. So we used a similar method of assessment by looking at animal behaviors, including spontaneous pain-related behaviors, produced by 10 stimulations with the 0.6g filament. We quantified nociceptive behaviors as following. 0; no withdrawal, 1; movements of the toes without withdrawal, 2; slow withdrawal, 3; fast withdrawal, 4; withdrawal with behaviors such as flinching and/or licking. Based on this scoring, we now reported a pain score in different experimental conditions:

- figure 4e, f: we evaluated paw withdrawal threshold and pain score after the activation of FLT3 via the intrathecal injection of FL. We reported that FL injection increased paw

withdrawal threshold and pain score, as well. Pain-related behaviors are completely blocked by shRNA FLT3 treatment.

- figure 6 g,h: we evaluated the effects of BDT001 in CCI animals and we show that BDT001 is effective in reducing both percentage of withdrawal threshold and pain score.

-Supplementary 4b, 6i: we show that Flt3 is specific in reducing pain-related behaviors associated with nerve injury since no effect was observed in inflammation-induced increased paw withdrawal threshold in *Flt3*^{KO} mice (supplementary figure 4b) or after BDT001 injection (figure 6i). Of note, similar results were obtained when we consider only the withdrawal response or the nociceptive behaviors. Another important point that reinforces the implication of Flt3 in nerve injury-induced pain hypersensitivity behavior is that we do not observe any difference in motor activity between the wild-type and Flt3 knock-out animals (see supplementary data Figure 2g) nor after chronic inflammatory pain in the CFA model (Supplementary Fig. 4b for paw withdrawal latency and figure below for pain score).

In addition, and most importantly, to further demonstrate the importance of Flt3 in the development of nerve injury-induced pain, we also considered the effects of Flt3 deletion in the conditioned place preference (CPP) paradigm. Indeed, the use of the classical nociceptive test (reflexive response or pain score) does not allow to evaluate spontaneous pain which is often more debilitating than the alteration of evoked nociceptive responses. To address this important issue, the conditioned place preference (CPP) paradigm has been developed to evaluate spontaneous ongoing pain in nonverbal animals, which depends on processing by limbic and cortical circuits and unlikely on spinal reflex. In the revised version of the manuscript we now include results obtained in *Flt3*^{KO} mice after CCI with the CPP procedure. The absence of clonidine-induced place preference in *Flt3*^{KO} mice strongly suggests that the deletion of *Flt3* reduces the aversive aspect of spontaneous pain. All the results obtained with the different behavioral tests go in the same way i.e. inhibition of Flt3 reduced pain-related behaviors associated with nerve injury. Altogether, these data strongly suggest the effects observed here are not due to modifications in spinal reflexes, but are due to alterations in nociceptive processing.

9. The data with the BDT001 compound is almost premature in this paper. There is a lot of characterization of this compound which could have been done in a separate manuscript. It is not clear to me what is the consequence of BDT001 inhibiting EphB4 receptor binding by 23%. This percent inhibition is not insignificant. How do the authors make sure the drug is selective to their target? Other Flt3 blockers certainly do have off-target effects. All this could be in a separate paper.

BDT001 has been profiled against 23 receptor tyrosine kinases (Suppl Table 3) and 49 intracellular kinases (Suppl Table 4). Apart from IRAK4 for which we found 64% inhibition at a concentration of 10 μ M, no other significant inhibition could be detected. The inhibition level of EphrinB2 binding to EphB4 (22.7%), still measured at a single concentration of 10 μ M of BDT001, is very weak, with respect to the herein demonstrated potency of the compound to inhibit neuronal FLT3 (IC₅₀ around 150 nM) and to the systemically circulating BDT001 concentration (190 nM) after administration of an in vivo active dose. According to the established standards of major pharmaceutical companies (see Bowes et al. Nat. Rev. Drug Discov., 2012, 11, 909-922), BDT001 can be considered as a selective FLT3 inhibitor with an off-target rate far below the acceptable threshold of 10%. Of course, we cannot rule out the possibility that this compounds binds to a still unknown target although the anti-hyperalgesic properties of BDT001 are clearly FLT3-dependent.

We respectfully disagree on this Reviewer's suggestion to exclude the data on this compound from the present paper. As stated by the other two referees (reviewer #1 and reviewer #2), we believe that developing an extracellular FLT3 inhibitor and providing

evidence for its in vivo antihyperalgesic properties in rodents gives a significant additional support to the proposed role of neuronal FL-FLT3 signaling in triggering neuropathic pain.

10. It is not clear to me what is the effect of BDT001 alone on the basal Ca level in DRGs.

We have now added this data to Fig. 6b showing that BDT001 has no effect on basal [Ca²⁺]_i levels in DRG neurons.

11. Putting in a mixture of techniques, including virtual screening, essentially makes reviewers unable to judge the whole manuscript. I am unable to tell whether the virtual screening was done using thorough methods and if proper controls were added or not. It is my opinion, nonetheless, that this compound should be more thoroughly validated elsewhere, in its own paper, before being incorporated here.

See above response to point 9. Please also note that we do not exclude to describe extensively in a specialized medicinal chemistry journal the screening methodology and structure-activity relationships obtained from novel chemical entities derived from BDT001.

Overall, the effect on mechanical pain sensing cannot really be interpreted because the appropriate test was not run. Even the effect on the Hargreaves' apparatus should be quantified in terms of the minimum latency to not only evoke a paw withdrawal, but a paw withdrawal followed by shaking or licking of the paw. Otherwise these behaviors remain spinal reflexes.

As requested by the reviewer, we evaluated the nociceptive behavior as requested by the reviewer after heat stimulation. We divided the latency of paw of withdrawal by a score which is as followed: 2; withdrawal, 3; sharp withdrawal with shaking or licking of the paw. The results are reported in the Figure 1d and below:

The role of neuronal Flt3 should be demonstrated. The expression of Flt3 should be confirmed not only by Western blot of DRGs, but also by showing that it is the neuronal Flt3 that is responsible for the effect.

See above response to point 1.

Minor comments:

Can the authors define what FL stands for?

FL stands for Flt3 Ligand (this has now been clarified in the manuscript)

What could control FLT3 expression level? One would think that nerve injury would upregulate FLT3, which would later be activated by FL. Why would neurons express FLT3 under normal conditions?

At the moment, we don't know what controls basal Flt3 levels. Our Q-PCR results show that nerve injury does not appear to affect Flt3 transcript levels. We do not think that FLT3 has to be upregulated after nerve injury, because it is exposure to its ligand that triggers the pathophysiological mechanism. We cannot speculate on why neurons express Flt3, since it would seem that it is not exposed to its ligand under normal conditions, and it does not seem to play a role in normal nociception. Could it be a detection system for peripheral nerve injury? Answering these interesting questions will require further work, outside the scope of the present article.

From the clinical point of view, because pharmacological blockade of FLT3 did not affect pain response in non-injured animals, a predicted therapeutic benefit of interfering with this receptor is that normal pain sensitivity would be unaffected.

Discussion page 8 line 289: FLT3 modulates the cytokine CSF1, ... What does that sentence mean? Modulates what aspect of CSF1? The expression level? The localization? Please be more precise.

Flt3 signaling modulates the expression of the cytokine Csf-1 in DRG neurons. This is now revised in the text (page 6, paragraph 1).

Introduction page 2 line 43: when describing the high social costs, it would be useful to give the readers an idea of what these numbers are. Readership is from mixed backgrounds and these are not obvious facts to everyone.

This is now specified (page 2, paragraph 1)

Page 2 line 55-57: this is a broad statement. Some references are necessary to back this claim.

References are now included (page 2, paragraph 2)

I have not heard the term chronicization. Most people in the field use chronification. For the sake of consistency, could you please use chronification?

We agree; this is now corrected.

Line 71L “secrete sensitizers”. This is another vague term being used. Give examples (with references) of molecules being released please, and avoid using vague terms.

We now put the terms “cytokines, chemokines and growth factors” to be more precise.

Page 3 line 83: the reference you want to cite is #16, not 10,11.

Thanks for noting the error. This is now corrected.

Results:

Based on the single cell PCR experiments of Usoskin et al., Nature Neuroscience (2015), Flt3 expression appears to be the highest in proprioceptors. Can the authors comment on this?

Our *in situ* hybridization results suggested a neuron-specific expression of Flt3 mRNA over a broad range of neuronal cell diameters in the DRG, that was absent in the null mutant. These in situ results were in broad agreement with the single cell experiments carried out on mouse DRG neurons by Usoskin et al., 2015, that showed expression of Flt3 in myelinated sensory neurons (proprioceptors, low threshold mechanoreceptors, A delta nociceptors) and at a minor level in a population of non peptidergic C fibers nociceptors. As highlighted by the reviewer the highest expression is found in proprioceptors. The role of this expression in proprioceptors was not directly investigated in our study, apart from the finding that in Flt3^{KO} mice the proportions of the neurons in well-described sensory sub-populations including the TRKC population were normal. In the same line of evidence, functional investigation of balance performance reveals that motor behaviors are similar between the wild-type and Flt3 null mutants. While an eventual compensation for the absence of FLT3 receptor expression cannot be excluded, our results suggest that FLT3 is not involved in the development or physiological functions of proprioceptive neurons.

Page 3 line 107: von Frey is with a small v

Corrected throughout the manuscript.

Page 4 line 147: "...recapitulates some of the molecular.."

This has been corrected

Page 6 line 206: it is not an "irrelevant shRNA", but rather a non-targeting shRNA. No shRNA is irrelevant.

This has been corrected throughout the manuscript.

Methods: define the anesthetic used (line 518)

The anesthetic used was isoflurane. It has been added in the method section.

Line 520: please edit the references, there is no reference #45

Reference #45 is listed in the online method paragraph in the "surgery" section

Line 544: the male mice argument should be in the first paragraph of methods.

Corrected

Line 556: the Dixon method: please provide the proper reference for this paper.

The reference for the Dixon method has been provided.

Dixon W. J. (1980) Efficient analysis of experimental observations. Annu. Rev. Pharmacol. Toxicol. 20, 441–462

Line 561: these tests are not measures of nociception.

Since we did not observe significant differences between withdrawal and nociceptive behaviors assessment, we kept the term nociceptive assay for the Von Frey, Acetone and Hargreave test.

Line 576: the cumulative reaction time: should be labeled simply as Duration of responses.

According to reviewer's comment, cumulative reaction time has been labeled as Duration of responses.

Line 578: it is not clear to me why anyone would inject 4 times capsaicin in the mouse paw. I can understand it is to mirror the in vitro data, but capsaicin produces quite a painful reaction in the mouse that can trigger central sensitization. Thus recruiting additional variables that have nothing to do with the processes going on in the sensory neuron.

We agree with the reviewer that repeated injections of capsaicin can produce pain sensitization. However, we did not observe significant enhancement of pain behavior in untreated FL animals suggesting that pain sensitization seems quite low at the doses used in our protocol. In addition, capsaicin injected twice with FL does not produce significant enhancement of pain-related behaviors and we need 3 injections to observe a significant potentiation in capsaicin response. The last injection of capsaicin was performed 24h after the last FL injection to determine whether the FL effects may produce sustained potentiation of capsaicin response.

Line 584: drug delivery. Please provide more information on catheterization: size, what is the catheter made of? Polyethylene? What is the internal diameter, outer diameter?; where was it anchored? Where was it inserted from? Where was is implanted? Text only says dorsal subcutaneous space.

More information has now been provided regarding the catheterization in the Material and Method section.

Line 592: a total of 10 ul of drug was injected. This seems quite high. If one was to inject 10 ul of lidocaine intrathecally, the animal would stop breathing because 10 ul can reach the brainstem. Doses used in mice are typically no more than 5 ul.

We would like to thank the reviewer for this note. However, in our conditions when we inject 10 µl intrathecally of an Evans blue solution for instance, we observe a staining reaching T11 thoracic level. In addition when we inject 10 µl of a solution of 21,33% lidocaine (Xylovet) we observe an impact motor block of the hindpaws and the animals are still breathing. Hence in our conditions, it looks like a volume of 10µl did not reach the brainstem.

Line 597: label as intra-sciatic injection. Was it the left or the right sciatic nerve bundle?

The intra-sciatic nerve injection was performed in the left sciatic nerve.

Gene knockdown experiments: siRNA approach: cells are overexpressing Flt3, thus at various levels. Using a knockdown approach like this makes it difficult to properly quantify how much knockdown you get. If the experiment cannot be done on the native cells (DRGs), then the authors should use a cell line endogenously expressing Flt3 and validate the siRNA knock down.

See Point 7 in major points.

Line 653: the cell dissociation method should be described before the electrophysiology or the Ca imaging.

We have now corrected this.

Line 685: what is Locks? Please define.

Locks is a standard external solution contained: 145 mM NaCl, 5 mM KCl, 2 mM CaCl₂, 2 mM MgCl₂, 10 mM HEPES, 10 mM glucose (pH adjusted to 7.4 with NaOH and osmolarity between 300 and 310 mOsm). It is now added in Materials and Methods "Calcium imaging" section.

Line 723: virus transfection: please organize better the justification of lines 724-731.

We can't do better; journal will take care of this.

Line 744: AAV not AVV

Corrected

Line 756: were41?

Corrected

Line 759: what is mowiol?

Mowiol is a mounting medium for microscopy. It's now revised in the text.

Virtual screening: seems premature to incorporate in this paper. The validation should be done elsewhere.

See major points 9)

Figures:

Figure 1c: you cannot use in the legend the term Mechanical Pain Sensitivity. Your test does not allow you to distinguish if the withdrawal was a reflex or if it was a nociceptive (not pain) response.

Since we did not observe important differences between withdrawal and nociceptive behaviors assessment, we kept the term mechanical pain sensitivity as classically used in the literature

Figure 1d. label should be duration of response, not reaction time. Reaction time may indicate the time it took for the mouse to respond. Similarly, "paw withdrawal threshold" is typically used for von Frey tests. "latency (sec)" is more appropriate here. Furthermore: the cold effect 1.5 sec to 2.5 sec is not a dramatic increase, despite being significant.

This has been changed as requested by the reviewer.

The heat latencies are quite low. The bulb intensity should be adjusted such that baseline values of ~10 seconds are needed. Then you would test the FL effect. You are near saturation latencies. Going from 5.5 sec to 4 sec is not dramatic, despite being significant.

We agree with the reviewer. This actually has been done for the figure 3b and for the supplementary data 2c. We usually adjust the baseline values around 8-10 sec. We faced some problem with the Hargreaves' test and we had to change bulb intensity. Despite being significant, as requested by the reviewer, we re-did the experiments by adjusting the baseline value around 10 sec.

Figure 2c: there is some additional information missing here. What happens to the Flt3 KO in response to capsaicin alone?

We have corrected this figure and added the responses to capsaicin alone of wild-type and Flt3 KO mice (Fig. 2c).

It seems the authors already have a patent on Flt3 receptor blockers in the treatment of pain. This makes me wonder if the data has been published elsewhere (patent).

The patent application (WO2011083124) does not contain any result of the present article, except a very small part of Fig. 2b.

References

1. Cavanaugh, D. J. *et al.* Restriction of transient receptor potential vanilloid-1 to the peptidergic subset of primary afferent neurons follows its developmental downregulation in nonpeptidergic neurons. *J. Neurosci. Off. J. Soc. Neurosci.* **31**, 10119–10127 (2011).
2. Hjerling-Leffler, J., Alqatari, M., Ernfors, P. & Koltzenburg, M. Emergence of functional sensory subtypes as defined by transient receptor potential channel expression. *J. Neurosci. Off. J. Soc. Neurosci.* **27**, 2435–2443 (2007).
3. Chung, M.-K., Güler, A. D. & Caterina, M. J. TRPV1 shows dynamic ionic selectivity during agonist stimulation. *Nat. Neurosci.* **11**, 555–564 (2008).
4. Mackarehtschian, K. *et al.* Targeted disruption of the flk2/flt3 gene leads to deficiencies in primitive hematopoietic progenitors. *Immunity* **3**, 147–161 (1995).
5. Ma, Q. P. & Woolf, C. J. Progressive tactile hypersensitivity: an inflammation-induced incremental increase in the excitability of the spinal cord. *Pain* **67**, 97–106 (1996).

Reviewers' comments:

Reviewer #1 (Remarks to the Author):

Reviewer #2 (Remarks to the Author):

1) The introduction states that "NP arises from aberrant functioning of somatosensory neurons present in DRG". This is incorrect because there are other forms of NP that do not involve DRG neurons. In addition, the animal models used here (CCI, SNI, SNL) solely represent NP associated with mechanical injury to peripheral nerves. This limits the conclusions because other (more clinically relevant) types of nerve injury (e.g. chemotherapy, infection, diabetes) are not represented by these models.

We have taken into consideration the reviewer's valuable criticism that there are other neuropathic pain conditions not caused by mechanical injury to peripheral neurons. Firstly, we now limit our conclusions to peripheral neuropathic pain (PNP instead of NP). Secondly, we now provide results obtained in a model of generalized PNP, induced by chemotherapy, showing that Flt3-deficient mice do not develop mechanical hypersensitivity after treatment with oxaliplatin (Fig 3g) and in Materials and Methods "Chronic pain models" section.

- Reviewer's comment: o.k.

2) The introduction also states that "Among [cytokines], with the notable exception of the cytokine FL and its cognate receptor FLT3, all the members of the class III receptor tyrosine kinase (RTK) family which comprises stem cell factor (SCF) receptor(c-Kit), colony-stimulating factor type-I (CSF1) receptor (CSF1R) and platelet-derived growth factor (PDGF) receptors (PDGFR) have been shown to be involved in normal nociception and/or pain". In view of the numerous non-immune factors involved in NP, this immediately raises the question whether the selective ablation of a single cytokine (FL) will ever be able to produce clinically relevant analgesia. This concern is further supported by their findings that "the capsaicin response was normal in cultures established from Flt3KO mice" (line 138), "BDT001 did not affect capsaicin-induced TRPV1 activation" (line 240), and "the compound at 10 μ M had no effect when TRPV1 potentiation was induced by NGF" (line 248). Again, this will limit the conclusions regarding clinical efficacy and indicates that a journal addressing a specialized audience may be more suitable for the rather narrow scope of this manuscript.

We would like to clarify these points. Our results do not concern "the selective ablation of a single cytokine (FL)" but the modulation of FLT3, its receptor. In addition, inhibiting FLT3 or ablating FLT3 expression does not, in any case, produce analgesia, but prevents or reverses hyperalgesia and pain sensitization after nerve lesion. Indeed, our results show that FLT3 has no role in normal nociception because:

- FLT3 stimulation does not activate nociceptors in physiological conditions (this is now specified in Fig. 2b, and text page 4, paragraph 3),

- Flt3-deficient mice have normal responses to capsaicin (now specified in Fig. 2c and text page 4, paragraph 3),

- Flt3-deficient mice have normal responses to mechanical, thermal (Supplementary Fig. 2c-f) or chemical nociception (new Supplementary Fig. 4a) and normal response to inflammatory pain (new Supplementary Fig. 4a, b),

- BDT001 does not affect normal nociception (Fig. 6j) nor inflammatory pain (Fig. 6i; new data).

However, peripheral neuronal FLT3 activation is necessary to induce PNP and its inhibition sufficient to reverse neuropathic hypersensitivity and many of the associated molecular changes. So FLT3 appears to be a key upstream modulator (trigger) of PNP physiopathology. Thus, we respectfully disagree with this Reviewer's skepticism that inhibition of FLT3, which is able to treat

neuropathic pain, will have clinical relevance. For these reasons we think that our findings are of general interest.

- Reviewer's comment: o.k., although it is still doubtful that the selective ablation of one single receptor (FLT3) will ever be of clinical relevance or of interest to the nonspecialized readership of this journal.

3) Numerous antibodies are used but control experiments to show their specificity are missing (with the exception of FLT3 in suppl. Fig. 2). This is a growing problem in the scientific literature and must be addressed (see e.g. Nature 2015;521:274).

The antibodies that were used in the present experiments are commercially available and have been widely used in similar experiments, for which there is an abundant literature. There are no discrepancies between the present data and those published regarding the proportions of identified neurons in WT animals (Supplementary Fig. 2a, b) or the distribution of markers in the DRG or DSC of CCI-WT animals (Fig 3i-l). The only antibody that had not been previously used was the anti-FLT3 antibody, and this antibody was validated for Western blotting by using Flt3-deficient mice (Supplementary Fig. 3c).

- Reviewer's comment: This concern was not addressed in a satisfactory manner. As discussed extensively in the community (e.g. Nature 2015;521:274), it is by no means sufficient to cite "abundant literature" that has used commercially available antibodies. Instead, it is necessary to demonstrate antibody specificity for each specific condition/tissue applied in the study at hand.

4) It is not clear what the authors mean by "gene" (e.g. lines 119, 121, 123, 285; Fig. 1). In the methods they describe RNA extraction for RT-PCR. Also, they state "gene knock down by siRNA" (line 602). How can siRNA influence a gene?

We have clarified these points in the text using the expressions "gene expression levels" or "gene transcripts" at many places in the manuscript.

- Reviewer's comment: o.k.

5) In Fig. 1c the authors showed an increased percentage of withdrawal 5 h after FL-injection. Did the authors check time points earlier than 5 h or is there any knowledge about the onset of FL-induced hypersensitivity and increased gene expression?

Fig. 1c shows clearly that FL-induced hyperalgesia is dose and time-dependent. Since at the highest dose tested (500 ng), the maximal effect was obtained 5 h after FL-injection (personal data), this time point was retained for subsequent experiments at the same dose.

- Reviewer's comment: The time- and dose-dependency of FL-induced hyperalgesia was not in doubt. The question was not about the time point of the maximal effect, but about the knowledge of the onset of FL-induced hypersensitivity and increased gene expression. This question was not answered.

6) Calcium imaging and patch clamp experiments in Fig. 2 show a rather rapid and direct way of action of FL, since the tachyphylaxis of TRP channels is reversed immediately after FL application. This cannot be explained by changes of expression levels, but possibly by phosphorylation events. Please comment on this.

We agree that such a short TRP activation by FL in cultured DRG neurons is likely to involve phosphorylation, as demonstrated for other TRP activators (see for example 3). This is now specified in the text. Nevertheless, as we show in Fig. 1g, FLT3 activation by FL is also likely to induce long-term effects through TRP gene upregulation.

- Reviewer's comment: This specification was not found in the text.

7) On page 4 line 135 the authors state that "FL alone had no effect on basal $[Ca^{2+}]_i$ ", however in

Fig. 2b the second column (only FL application (0.56 nM) shows almost the same ΔF peak as capsaicin alone which was shown to induce a calcium influx. This is contradictory, please comment on this. Additionally, it is not clear to which capsaicin application the response amplitudes were normalized (Peak 1 could probably be excluded).

The Reviewer is correct; this was an omission. Figure 2b has now been corrected and shows that FL alone has no effect on basal $[Ca^{2+}]_i$. Pulses of capsaicin were applied 4 times at 2 min-intervals and capsaicin with drugs was added after the third pulse. The normalization was done by calculating the ratio of Peak4/Peak3. This now added in the Material and Methods "Calcium imaging" section.

- Reviewer's comment: The newly presented data are inconsistent. Why is the capsaicin concentration for calcium imaging experiments now 2.5 μM (see legend for Fig. 2a+b+d)? In the original version of the manuscript, the capsaicin concentration was 0.5 μM . Which capsaicin concentration was used for calcium imaging experiments?

8) The finding that "the capsaicin response was normal in cultures established from Flt3KO mice" (line 138) is in contrast to "The mRNA expression of ... TrpV1 ... in DRG ... were reduced after Flt3 deletion" (line 174). Please comment.

This was poorly written. In fact, Fig. 3h shows that there was no difference in TrpV1 gene expression between WT and KO mice in non-injured animals, from which DRG cultures were established; it is therefore not surprising that the capsaicin response was not altered. However, increases in mRNA expression of Atf3, NpY, TrpV1 and Gfap in DRG and Iba1, Cd11b and Gfap in DSC were abrogated or diminished after Flt3 deletion in nerve injured animals. The text has been corrected accordingly (page 6, paragraph 1).

- Reviewer's comment: The concerns are addressed. However, in Fig. 3h there is no injury-induced increase of Gfap in DSC in CCI wt mice which could be reversed (at least there is no statistical difference indicated in the figure).

9) Please describe the stimulation protocol for TRPA1 and TRPM8. Was it the same as for TRPV1? The same temporal stimulation protocol was used for the three TRPs. This is now added to the Materials & Methods "Calcium imaging" section.

- Reviewer's comment: o.k.

10) Why did the authors use different concentrations of capsaicin and FL in their patch clamp experiments?

First, many previously published patch-clamp experiments have used a range of concentrations between 0.5-1 μM to induce capsaicin currents in current clamp experiments, with better kinetics at 1 μM . Second, we wanted to have saturating concentrations of FL (between 5.6 and 10 nM) to analyze this effect with dynamic kinetics that is more rapid with patch clamp than with calcium imaging.

- Reviewer's comment: o.k.

11) Please also state the number of animals from which the cells were taken for measurements in Fig. 2b-f.

Calcium imaging have been done for at least 3 different animals for each experimental conditions. This is now specified in Materials and Methods, "Calcium imaging" section, page 18.

- Reviewer's comment: There is no specification about the number of animals on page 18 (neuron culture and patch clamp experiments) or 19 (calcium imaging experiments). This information is still missing in the manuscript.

12) Suppl. Fig. 2b: Did the authors also check other FL concentrations? Possibly the missing effect

was due to a wrong dose? It is not clear for the reviewer why systemic FL should not work like near nerve injection (unless FL is not able to permeate the blood vessel wall), especially since it was shown in Fig. 6d that systemic intrathecal FL injection induced mechanical pain hypersensitivity.

We have repeated this experiment using a dose of 5 mg/100 μ l for the intravenous injection. The result was again negative with this high dose. This is now revised in the new Supp. Fig. 3b.

- Reviewer's comment: In Supp. Fig. 3b the dose is stated as "5 μ g/100 μ l" not "5 mg/100 μ l". Which dose is correct?

13) why systemic FL should not work like near-nerve injection?

FL nerve injection is done to bypass the peripheral nerve-blood barrier. It is not a "near nerve injection", but an "intra-nerve injection". In Fig. 6d it was not "systemic" intrathecal injection, but localized intrathecal injection. This intrathecal injection applies the solution into subarachnoid space which is contiguous with the dorsal roots.

- Reviewer's comment: o.k.

14) Suppl. Fig. 2c: Please provide also the appropriate controls for this experiment.

We have added a positive control to the Western blot, showing the detection of FLT3 protein in one brain tissue i.e. cerebellum and the absence of the FLT3 band in Flt3-null mutant tissue (Supplementary Fig. 3c).

- Reviewer's comment: o.k.

15) For consistency of data presentation in Fig. 3 it would be good to show the time course for Fig. 3b too, as it was done for the other behavioral tests in this Fig. However this is a minor point. We have not performed a complete time-course. To limit the number of animals used in our investigation, we performed the Hargreaves test and the von Frey test in the same groups of animals. Hence, the repetition of nociceptive tests in nerve injured-animals was limited here to avoid a potential pain sensitization produced by the repeated application of the nociceptive stimulus itself.

- Reviewer's comment: This is a major limitation for the interpretation of results. The repeated application of painful stimuli (of any nature) to the same animal is likely to produce either sensitization or tolerance, i.e. the animals will either show enhanced reactions because they "anticipate" pain or they "get used" to painful stimuli and therefore show reduced reactions. This limitation must be elaborated in the discussion. Alternatively, the results of the second test may be removed from the manuscript.

16) Why did the authors choose the two different time points (3 days post CCI and 14 days post CCI) to investigate mRNA expression and immunoreactivity for neuropathic pain markers?

In mice, the first week corresponds to the period of development of the peripheral and central hypersensitivity and the installation of the PNP pathology. Thereafter, the pathological state is stable and maintained (see Fig. 3a; Fig. 4a,g). By choosing these two time points, we can therefore assess both the changes during the dynamic process and the stable state.

- Reviewer's comment: o.k.

17) Page 5 lines 181 and 182: The authors state that NP-induced upregulation of mRNA of Csf1 ligand and CsfR is diminished in Flt3 KO mice. However, injection of CSF1 restored mechanical hyperalgesia in KO mice. How does this work if Csf1 receptor is downregulated? Please comment on this.

Our description was not clear. Fig. 3m shows that the injury-induced increases of CSF1 and CSF1R expression were attenuated in Flt3 null mutant mice, but CSF1 and CSF1R were normal in sham-

operated Flt3 KO mice. So, this is not surprising that Flt3 KO mice respond to CSF1 in Fig. 3p. The text has been modified accordingly (page 6, paragraph 1).

- Reviewer's comment: o.k.

18) Page 6 lines 202 and 203: The authors state that "GFP expression was restricted to DRG neurons." But in Suppl Fig. 4a and b the expression of anti-Flt3 shRNA and GFP is shown in DSC. Please explain the discrepancy.

AAV9 cell transduction, as reported by GFP, is limited to neurons in the DRG (co-staining with Tuj1) and was excluded from other cellular populations of the DRG such as CD45+ resident macrophages and GFAP+ satellite glial cells (Fig. 4d). GFP staining in the dorsal spinal cord labeled fiber-like processes, which almost certainly originate from sensory afferent projections, and was absent from NeuN+ neuron cell bodies and CD45+ microglia (Suppl. Fig. 6a, b).

- Reviewer's comment: o.k.

19) Please state the number experiments for Suppl. Fig. 5.
It is now in Suppl. Fig. 6. and the experiments were done in triplicate.

- Reviewer's comment: o.k.

20) Page 6 lines 234-236: It is stated that "BDT001 inhibited FL-induced FLT3 phosphorylation in leukemia-derived RS4-11 cells, also measured by TR-FRET (Supplementary Fig. 5)...". However in Suppl. Fig. 5 BDT001 was not investigated but another inhibitor AC220. Please comment on that. In this Figure, now Suppl Fig.7, we describe the two in vitro assays (panel a: binding of FL to FLT3 in HEK cells, panel c: FL-induced phosphorylation of FLT3 in RS4-11 cells) that we have set-up to evidence extracellular FLT3 inhibition by competitive binding (panel b) and blockade of FL-induced FLT3 phosphorylation (panel d). Since no extracellular FLT3 inhibitor was known when the assay was developed, validation of the phosphorylation assay was done using an intracellular non-selective FLT3 inhibitor (AC220). Once the assay validated (Suppl Fig. 7), it has been later used to screen potential extracellular FL3 inhibitors and identify BDT001 (Fig. 5d-g).

- Reviewer's comment: This is still not clear. On page 7 lines 257-259 it is stated "One compound (compound 3, Fig. 5d, e) effectively prevented extracellular FL binding to FLT3, as measured by time-resolved fluorescence resonance energy transfer (trFRET, Supplementary Fig. 7)." Compound 3 is BDT001, therefore this sentence reads that BDT001 prevented extracellular FL binding to FLT3, as measured by trFRET (Suppl. Fig. 7). But if one looks at Suppl. Fig. 7 there are NO data of BDT001, but AC220. Why do the authors refer to a figure which is unrelated to BDT001? The explanation about the validation of the assay does not answer this question. The correct reference for this statement would be Fig. 5g or the sentence should be changed to "One compound (compound 3, Fig. 5d, e) effectively prevented extracellular FL binding to FLT3, as measured by time-resolved fluorescence resonance energy transfer (Fig. 5g)."

21) Please discuss long-term effects of FLT3 inhibition as it was shown to be involved in proliferation and differentiation of hematopoietic cells. Is there any influence on lymphocyte development?

Flt3-deficient mice have no gross abnormalities in their hematopoietic system: they have normal absolute numbers of mature myeloid and lymphoid cells and thymocytes subpopulations in the spleen and of mature monocyte, granulocyte and erythrocyte numbers in the bone marrow; however, they show a 33% reduction of early B -cell progenitors, whereas the number of more mature B cells is normal. We summarize these literature findings in the text by stating that Flt3-deficient mice have a normal mature hematopoietic system, even though some discrete populations of cell progenitors are reduced (page 3, paragraph 2).

- Reviewer's comment: o.k.

21) The reference list is divided into 2 parts by the method section (Ref. 1-43 on pages 10-13; Refs. 44-52 on pages 22-23).

It is now corrected

- Reviewer's comment: o.k.

22) In the method section describing calcium imaging experiments it is stated that DRG neurons were loaded with 5 mM Fura-2 AM. This Fura concentration seems to be very high. Usually concentrations between 3 and 5 μ M are used in the literature.

This was a mistake, now corrected: the concentration was 2.5 μ M.

- Reviewer's comment: o.k.

23) Page 22, line 834/835: It is stated that 2.9 commercially available drug-like compounds were filtered. The authors probably meant 2.9 million compounds, since this is stated in Fig. 5c. We acknowledge the Reviewer for pinpointing this discrepancy. Indeed, 2.9 million compounds have been screened in silico. The manuscript has been modified accordingly page 22 in the 'Virtual screening' section of the 'Online Methods' section.

- Reviewer's comment: o.k.

24) Suppl. Fig. 7, page 7 lines 182-184: "The ortho-phenol moiety seems to be mandatory for FLT3 binding since its deletion (compounds 68, 74, 77), displacement to the meta position (compound(s) 62), or methylation (compound 69) increases IC50 values. However, compound 69 displays the 3rd smallest IC50 value of all investigated compounds, right after compounds 66 and 75 (mentioned on page 6 being highly effective in inhibiting FL binding). Why do the authors conclude an increase of IC50 value for compound 69?"

The reviewer is right in stating that the methoxy analog 69 is among the best binders. However, it only partially inhibits FLT3 (35% inhibition at 100 μ M) with respect to the phenol derivative 11 (66% inhibition 100 μ M). To ascertain the deleterious effect of methylating the phenol group, we have synthesized and tested the methoxy analog of BDT001. Its much weaker potency (IC50 = 58 μ M vs. IC50=17 μ M for BDT001) supports our original statement.

- Reviewer's comment: o.k.

25) The grammar and style of the supplementary methods is poor. It is hard to follow and there is an inconsistency in spelling (μ L vs μ l, room temperature vs Room temperature, simple present passive vs simple past passive...), spelling errors and misleading verbs. Please revise this section. The manuscript has been fully edited by a native English speaker (Patrick Carroll). Modifications have been carried out throughout the text.

- Reviewer's comment: o.k.

Reviewer #3 (Remarks to the Author):

most of my comments have been addressed. Just a few remaining issues, which shouldn't necessitate additional experiments.

On the issue of why are the effects of FL injection measured only 5 hours post injection? Why not test the mice earlier?

The answer provided by the authors is weak. I have no problem with the fact that they say it doesn't have much effect when tested before 5 hours, and that they have personal data about it, but then they should show it. Or give the data in the text. One doesn't simply inject intrathecally a sensitizing compound to show that its effect only shows up 5 hours later. This is important data relating to the kinetics of the cytokines effect.

On the issue of: "A MAJOR concern of the methodology that is of concern is the test used to assess mechanical nociception. (Fig. 4 a and Fig S3c)..."

the answer/approach taken by the authors is fine. Their response must have a mistake in it as they claim (end of page 13 of rebuttal): "we reported that FL injection increased paw withdrawal threshold and pain score, as well". Presumably they mean REDUCED paw withdrawal threshold... or elevated percentage of withdrawals...

Just verify this in the text to be sure of the direction that the data goes.

Rest of the manuscript is fine

Responses to reviewers

Reviewers' comments:

Reviewer #1 (Remarks to the Author):

Reviewer #2 (Remarks to the Author):

3) Numerous antibodies are used but control experiments to show their specificity are missing (with the exception of FLT3 in suppl. Fig. 2). This is a growing problem in the scientific literature and must be addressed (see e.g. Nature 2015;521:274).

Authors' comments: *The antibodies that were used in the present experiments are commercially available and have been widely used in similar experiments, for which there is an abundant literature. There are no discrepancies between the present data and those published regarding the proportions of identified neurons in WT animals (Supplementary Fig. 2a, b) or the distribution of markers in the DRG or DSC of CCI-WT animals (Fig 3i-l). The only antibody that had not been previously used was the anti-FLT3 antibody, and this antibody was validated for Western blotting by using Flt3-deficient mice (Supplementary Fig. 3c).*

- **Reviewer's comment:** This concern was not addressed in a satisfactory manner. As discussed extensively in the community (e.g. Nature 2015;521:274), it is by no means sufficient to cite "abundant literature" that has used commercially available antibodies. Instead, it is necessary to demonstrate antibody specificity for each specific condition/tissue applied in the study at hand.

We acknowledge that the use of low-quality antibodies or lack of validation is a serious problem in the scientific literature. The quoted *Nature* paper (2015, 521: 274) highlights the problem, and provide a few recommendations on how to perform validation with positive and negative controls. To look at how these recommendations have been taken into considerations, we made a survey on Articles and Letters, which have used antibodies in various applications, recently published in *Nature* (from Oct 5, 2017 to Nov 9, 2017). The survey includes examination of the Life Sciences Reporting Summary, which is published online at this journal's website. Of 20 papers scrutinized, 8 papers list only the manufacturer and catalog number. Three papers list manufacturer, catalog number and previous literature. One paper lists manufacturer, catalog number and add that the antibodies were home-validated, but does not give any details on the validation. Six papers add to information above, that the antibodies have been validated by manufacturers; some include website links to the manufacturer's data sheets. We found one paper referring to Antibodypedia. We found only 4 papers that, in addition to information as above, have used true negative controls (gene inactivation, transfection with empty vector, knockdown with siRNA, knockout mice), and this for only one the antibodies used in the study.

In the new revised version, we have added a subsection in Materials and Methods "Antibodies", which complies with what appears to be the *Nature's* highest standard. All antibodies in the study, except the anti-FLT3 antibody, are very commonly used as phenotypical markers for glial cells or neurons, for

which the pattern of immunostaining distribution is well known. We have added previous literature from our lab utilizing these antibodies in these experimental conditions. We are specifying that for each antibody, a control has been made to check in our lab conditions that the distribution of cell count was identical to what is well known from an abundant previous literature. For the anti-FLT3 antibody, we are specifying that this antibody has been validated with *Flt3* knockout mice. For each antibody, we are providing the manufacturer and catalog number, link to the manufacturer's data sheet, which contains validation data.

For several antibodies (ATF3, NP-Y, GFAP, Iba1, CSF1) the results of IMH were reinforced by quantification of mRNA levels in the tissue using Q-PCR (e.g. Fig. 1g; Fig. 3h), providing an additional level of validation.

5) In Fig. 1c the authors showed an increased percentage of withdrawal 5 h after FL-injection. Did the authors check time points earlier than 5 h or is there any knowledge about the onset of FL-induced hypersensitivity and increased gene expression?

Authors' comment: *Fig. 1c shows clearly that FL-induced hyperalgesia is dose and time-dependent. Since at the highest dose tested (500 ng), the maximal effect was obtained 5 h after FL-injection (personal data), this time point was retained for subsequent experiments at the same dose.*

- **Reviewer's comment:** The time- and dose-dependency of FL-induced hyperalgesia was not in doubt. The question was not about the time point of the maximal effect, but about the knowledge of the onset of FL-induced hypersensitivity and increased gene expression. This question was not answered.

As requested by the reviewer, we conducted additional experiments to determine the onset of FL-induced hyperalgesia by measuring paw withdrawal threshold 30, 60, 90 and 120 minutes after intrathecal injection of the highest dose of FL used in our study (500ng). In our conditions, we observed a significant increase in the percentage of withdrawal starting at 2h post-FL injection, but not at shorter time-points. This information is now indicated in the manuscript in the results section, 2nd paragraph. We have measured changes in gene expression at maximal behavioral effects i.e. 24h after FL injection and we did not perform any time course.

6) Calcium imaging and patch clamp experiments in Fig. 2 show a rather rapid and direct way of action of FL, since the tachyphylaxis of TRP channels is reversed immediately after FL application. This cannot be explained by changes of expression levels, but possibly by phosphorylation events. Please comment on this.

Authors' comment: *We agree that such a short TRP activation by FL in cultured DRG neurons is likely to involve phosphorylation, as demonstrated for other TRP activators (see for example 3). This is now specified in the text. Nevertheless, as we show in Fig. 1g, FLT3 activation by FL is also likely to induce long-term effects through TRP gene upregulation.*

- **Reviewer's comment:** This specification was not found in the text.

We apologize for not addressing this issue. Phosphorylation of TRPV1 as a likely mechanism of rapid modulation is now specified in Discussion, first paragraph, with a supporting reference.

7) On page 4 line 135 the authors state that "FL alone had no effect on basal $[Ca^{2+}]_i$ ", however in Fig. 2b the second column (only FL application (0.56 nM) shows almost the same ΔF peak as capsaicin

alone which was shown to induce a calcium influx. This is contradictory, please comment on this. Additionally, it is not clear to which capsaicin application the response amplitudes were normalized (Peak 1 could probably be excluded).

Authors' comment: *The Reviewer is correct; this was an omission. Figure 2b has now been corrected and shows that FL alone has no effect on basal $[Ca^{2+}]_i$. Pulses of capsaicin were applied 4 times at 2 min-intervals and capsaicin with drugs was added after the third pulse. The normalization was done by calculating the ratio of Peak4/Peak3. This now added in the Material and Methods "Calcium imaging" section.*

- **Reviewer's comment:** The newly presented data are inconsistent. Why is the capsaicin concentration for calcium imaging experiments now 2.5 μ M (see legend for Fig. 2a+b+d)? In the original version of the manuscript, the capsaicin concentration was 0.5 μ M. Which capsaicin concentration was used for calcium imaging experiments?

We apologize for this mistake. The capsaicin concentration is 0.5 μ M. This was corrected in legends (Fig. 2)

8) The finding that "the capsaicin response was normal in cultures established from Flt3KO mice" (line 138) is in contrast to "The mRNA expression of ... TrpV1 ... in DRG ... were reduced after Flt3 deletion" (line 174). Please comment.

Authors' comment: *This was poorly written. In fact, Fig. 3h shows that there was no difference in TrpV1 gene expression between WT and KO mice in non-injured animals, from which DRG cultures were established; it is therefore not surprising that the capsaicin response was not altered. However, increases in mRNA expression of Atf3, NpY, TrpV1 and Gfap in DRG and Iba1, Cd11b and Gfap in DSC were abrogated or diminished after Flt3 deletion in nerve injured animals. The text has been corrected accordingly (page 6, paragraph 1).*

- **Reviewer's comment:** **The concerns are addressed.** However, in Fig. 3h there is no injury-induced increase of Gfap in DSC in CCI wt mice which could be reversed (at least there is no statistical difference indicated in the figure).

We agree with the reviewer on this detail. In consequence, we removed the claim that Gfap mRNA was reduced in the DSC: in the text, it is now written as "The injury-induced increases in transcripts of Atf3, NpY, TrpV1 and Gfap in DRG and Iba1, Cd11b in DSC were abrogated or diminished after Flt3 deletion, at 3 days post-CCI (Fig. 3h)."

11) Please also state the number of animals from which the cells were taken for measurements in Fig. 2b-f.

Authors' comment: *Calcium imaging have been done for at least 3 different animals for each experimental conditions. This is now specified in Materials and Methods, "Calcium imaging" section, page 18.*

- **Reviewer's comment:** There is no specification about the number of animals on page 18 (neuron culture and patch clamp experiments) or 19 (calcium imaging experiments). This information is still missing in the manuscript.

We used 6 animals for path-clamp experiments and 21 animals for calcium imaging experiments. This was added in Materials and Methods, section "Adult sensory neuron culture".

12) Suppl. Fig. 2b: Did the authors also check other FL concentrations? Possibly the missing effect was due to a wrong dose? It is not clear for the reviewer why systemic FL should not work like near nerve injection (unless FL is not able to permeate the blood vessel wall), especially since it was shown in Fig. 6d that systemic intrathecal FL injection induced mechanical pain hypersensitivity.

Authors' comment: *We have repeated this experiment using a dose of 5 mg/100 μ l for the intravenous injection. The result was again negative with this high dose. This is now revised in the new Supp. Fig. 3b.*

- **Reviewer's comment:** In Supp. Fig. 3b the dose is stated as "5 μ g/100 μ l" not "5 mg/100 μ l". Which dose is correct?

We apologize for the mistake. The dose of FL used is 5 μ g/100 μ l as stated in Supp. Fig. 3b

15) For consistency of data presentation in Fig. 3 it would be good to show the time course for Fig. 3b too, as it was done for the other behavioral tests in this Fig. However this is a minor point.

Authors' comment: *We have not performed a complete time-course. To limit the number of animals used in our investigation, we performed the Hargreaves test and the von Frey test in the same groups of animals. Hence, the repetition of nociceptive tests in nerve injured-animals was limited here to avoid a potential pain sensitization produced by the repeated application of the nociceptive stimulus itself.*

- **Reviewer's comment:** This is a major limitation for the interpretation of results. The repeated application of painful stimuli (of any nature) to the same animal is likely to produce either sensitization or tolerance, i.e. the animals will either show enhanced reactions because they "anticipate" pain or they "get used" to painful stimuli and therefore show reduced reactions. This limitation must be elaborated in the discussion. Alternatively, the results of the second test may be removed from the manuscript.

As we stated in our first response, and in agreement with the reviewer, we limited the number of stimuli on the same animals to avoid sensitization or tolerance. Sensitization or habituation usually appears when the tests are performed in a short interval of time. In fig 3a, we performed von Frey test only twice a week during the first two weeks and then once a week until the end of the experiment. Since the paw withdrawal threshold was maximal at one week and remained almost unchanged thereafter, we assume that there was no sensitization or tolerance. Also to avoid repetition of painful stimuli, we performed a single Hargreave's test at Day 14 post-CCI (Fig. 3b), on the same animals used in Fig. 3a. Since we show that, contrary to CCI-WT mice, CCI-Flt3 KO mice did not show any change in response in the Hargreave's test, our conclusion cannot be confounded by sensitization or tolerance.

We added this sentence in section Results, "Neuronal FLT3 is critical for development and maintenance of PNP", first paragraph : **"Note that the repetition of painful mechanical stimulus applications is unlikely to produce sensitization or tolerance since the maximal change in paw withdrawal threshold was achieved after 2 weeks and was maintained over time until the end of the experiment"**.

20) Page 6 lines 234-236: It is stated that “BDT001 inhibited FL-induced FLT3 phosphorylation in leukemia-derived RS4-11 cells, also measured by TR-FRET (Supplementary Fig. 5)...”. However in Suppl. Fig. 5 BDT001 was not investigated but another inhibitor AC220. Please comment on that.

Authors' comment: *In this Figure, now Suppl Fig.7, we describe the two in vitro assays (panel a: binding of FL to FLT3 in HEK cells, panel c: FL-induced phosphorylation of FLT3 in RS4-11 cells) that we have set-up to evidence extracellular FLT3 inhibition by competitive binding (panel b) and blockade of FL-induced FLT3 phosphorylation (panel d). Since no extracellular FLT3 inhibitor was known when the assay was developed, validation of the phosphorylation assay was done using an intracellular non-selective FLT3 inhibitor (AC220). Once the assay validated (Suppl Fig. 7), it has been later used to screen potential extracellular FL3 inhibitors and identify BDT001 (Fig. 5d-g).*

- **Reviewer's comment:** This is still not clear. On page 7 lines 257-259 it is stated “One compound (compound 3, Fig. 5d, e) effectively prevented extracellular FL binding to FLT3, as measured by time-resolved fluorescence resonance energy transfer (trFRET, Supplementary Fig. 7).” Compound 3 is BDT001, therefore this sentence reads that BDT001 prevented extracellular FL binding to FLT3, as measured by trFRET (Suppl. Fig. 7). But if one looks at Suppl. Fig. 7 there are NO data of BDT001, but AC220. Why do the authors refer to a figure which is unrelated to BDT001? The explanation about the validation of the assay does not answer this question. The correct reference for this statement would be Fig. 5g or the sentence should be changed to “One compound (compound 3, Fig. 5d, e) effectively prevented extracellular FL binding to FLT3, as measured by time-resolved fluorescence resonance energy transfer (Fig. 5g).”

We apologize for the misunderstanding. As was specified in the text and Fig 5e, compound #3 is not BDT001 (see Fig. 5e), but an analog. The first sentence refers to identification of compound #3, as an inhibitor of FL binding (Fig. 5d), as measured by trFRET, for which the assay validation is given on Suppl. Fig. 7a, b: FL saturation and inhibition of FL binding by FL). Then, the data for BDT001, a compound derived from compound Hit #3, are provided for inhibition of FL binding (measured on FL saturation at several BDT001 concentrations, Fig. 5f) and for inhibition of FL-induced phosphorylation (measured at several FL and BDT001 concentrations, Fig. 5g). FLT3 phosphorylation is also measured by trFRET, for which the assay validation is provided on Suppl. Fig. 7c, d: concentration-response for FL stimulation and inhibition by an FLT3 inhibitor). Because there is no known extracellular FLT3 inhibitor, we used a well-known intracellular inhibitor, AC220, to validate the phosphorylation assay. The text has been altered to be read more easily.

Reviewer #3 (Remarks to the Author):

Most of my comments have been addressed. Just a few remaining issues, which shouldn't necessitate additional experiments.

On the issue of why are the effects of FL injection measured only 5 hours post injection? Why not test the mice earlier?

The answer provided by the authors is weak. I have no problem with the fact that they say it doesn't have much effect when tested before 5 hours, and that they have personal data about it, but then

they should show it. **Or give the data in the text.** One doesn't simply inject intrathecally a sensitizing compound to show that its effect only shows up 5 hours later. This is important data relating to the kinetics of the cytokines effect.

As requested by the reviewer, we conducted additional experiments to determine the onset of FL-induced hyperalgesia by measuring paw withdrawal threshold 30, 60, 90 and 120 minutes after intrathecal injection of the highest dose of FL used in our study (500ng). In our conditions, we observed a significant increase in the percentage of withdrawal starting at 2h post-FL injection. This information is now indicated in the manuscript in the results section, 2nd paragraph.

On the issue of: "A MAJOR concern of the methodology that is of concern is the test used to assess mechanical nociception. (Fig. 4 a and Fig S3c)...". The answer/approach taken by the authors is fine. Their response must have a mistake in it as they claim (end of page 13 of rebuttal): "we reported that FL injection increased paw withdrawal threshold and pain score, as well". Presumably they mean REDUCED paw withdrawal threshold... or elevated percentage of withdrawals...Just verify this in the text to be sure of the direction that the data goes.

The referee is right. This was corrected in Results section, paragraph 2 (Fig. 1c).

Rest of the manuscript is fine

REVIEWERS' COMMENTS:

Reviewer #2 (Remarks to the Author):

The responses and respective revisions are now acceptable. We do advise to evaluate the authors' answers regarding antibody specificity from an editorial point of view.

Reviewer #3 (Remarks to the Author):

MY comments were almost completely addressed in the previous version. They are addressed in this version.

Responses to reviewers

Reviewers comments:

Reviewer #2 (Remarks to the Author):

The responses and respective revisions are now acceptable. We do advise to evaluate the authors' answers regarding antibody specificity from an editorial point of view.

Reviewer #3 (Remarks to the Author):

My comments were almost completely addressed in the previous version. They are addressed in this version.

We would like to thank these Reviewers for their rigorous evaluation of our study and their suggestions that have improved the clarity of the manuscript.